# Vidu4D: Single Generated Video to High-Fidelity 4D Reconstruction with Dynamic Gaussian Surfels

**Yikai Wang**[*1], **Xinzhou Wang**[*1,2,3], **Zilong Chen**[1,2], **Zhengyi Wang**[1,2], **Fuchun Sun**[1], **Jun Zhu**[†1,2]

[1]Department of Computer Science and Technology, BNRist Center, Tsinghua University
[2]ShengShu    [3]College of Electronic and Information Engineering, Tongji University
`yikaiw@outlook.com, wangxinzhou@tongji.edu.cn, dcszj@tsinghua.edu.cn`

## Abstract

Video generative models are receiving particular attention given their ability to generate realistic and imaginative frames. Besides, these models are also observed to exhibit strong 3D consistency, significantly enhancing their potential to act as world simulators. In this work, we present Vidu4D, a novel reconstruction model that excels in accurately reconstructing 4D (*i.e.*, sequential 3D) representations from single generated videos, addressing challenges associated with non-rigidity and frame distortion. This capability is pivotal for creating high-fidelity virtual contents that maintain both spatial and temporal coherence. At the core of Vidu4D is our proposed *Dynamic Gaussian Surfels* (DGS) technique. DGS optimizes time-varying warping functions to transform Gaussian surfels (surface elements) from a static state to a dynamically warped state. This transformation enables a precise depiction of motion and deformation over time. To preserve the structural integrity of surface-aligned Gaussian surfels, we design the warped-state geometric regularization based on continuous warping fields for estimating normals. Additionally, we learn refinements on rotation and scaling parameters of Gaussian surfels, which greatly alleviates texture flickering during the warping process and enhances the capture of fine-grained appearance details. Vidu4D also contains a novel initialization state that provides a proper start for the warping fields in DGS. Equipping Vidu4D with an existing video generative model, the overall framework demonstrates high-fidelity text-to-4D generation in both appearance and geometry. Project page: `https://vidu4d-dgs.github.io`.

## 1 Introduction

The field of multimodal generation exhibits significant advancements and holds great promise for various applications. Recently, video generative models have garnered attention for their remarkable capability to craft immersive and lifelike frames [4, 8]. These models produce visually stunning content while also exhibiting strong 3D consistency [15, 81], largely increasing their potential to simulate realistic environments.

Parallel to these developments, high-quality 4D reconstruction has made great strides [19, 58, 63, 94, 100]. This technique involves capturing and rendering detailed spatial and temporal information. When integrated with generative video technologies, 4D reconstruction potentially enables the creation of models that capture static scenes and dynamic sequences over time. This synthesis provides a more holistic representation of reality, which is crucial for applications such as virtual reality, scientific visualization, and embodied artificial intelligence.

---

[*]Equal contribution.    [†]The corresponding author.

38th Conference on Neural Information Processing Systems (NeurIPS 2024).

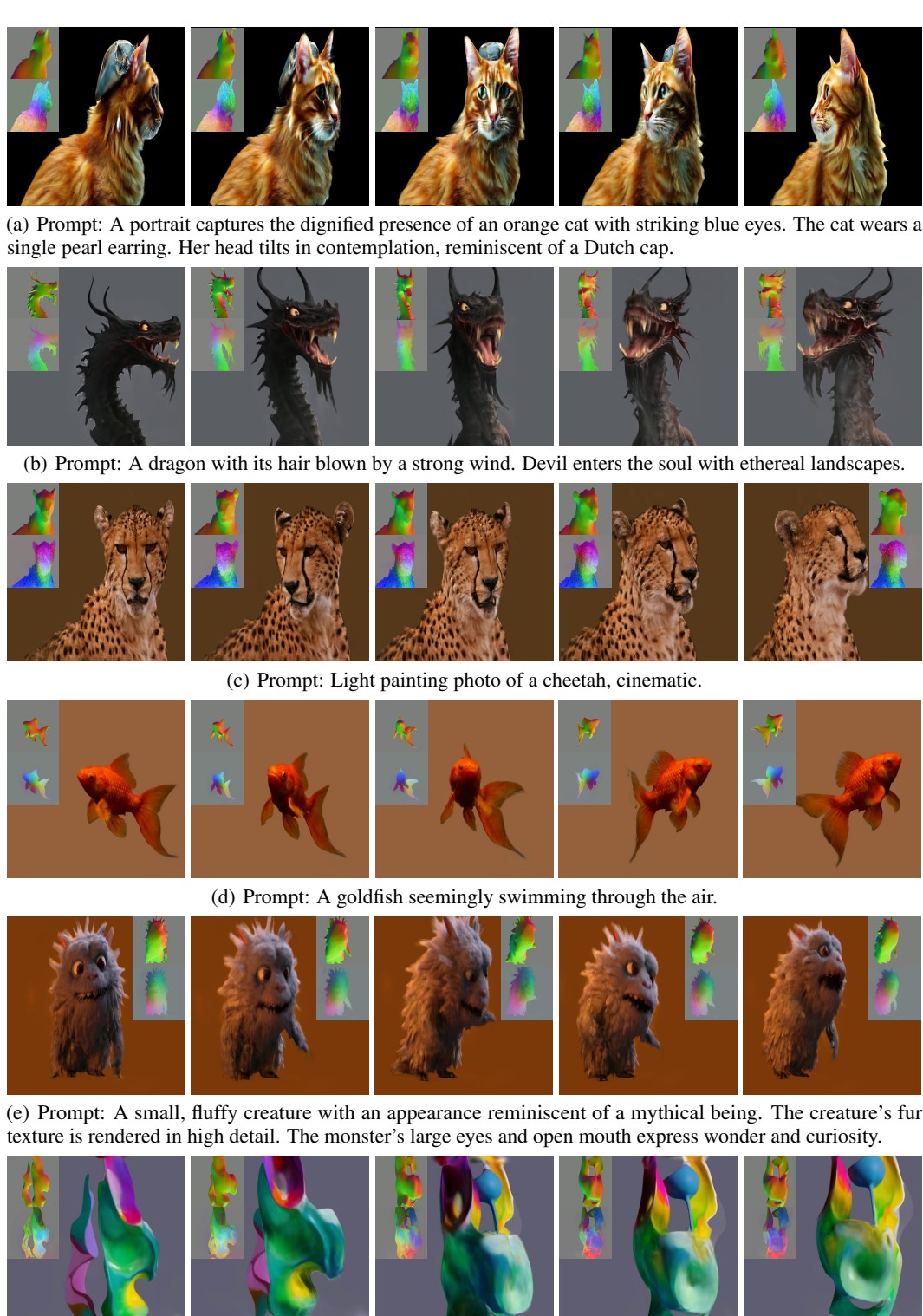

(a) Prompt: A portrait captures the dignified presence of an orange cat with striking blue eyes. The cat wears a single pearl earring. Her head tilts in contemplation, reminiscent of a Dutch cap.

(b) Prompt: A dragon with its hair blown by a strong wind. Devil enters the soul with ethereal landscapes.

(c) Prompt: Light painting photo of a cheetah, cinematic.

(d) Prompt: A goldfish seemingly swimming through the air.

(e) Prompt: A small, fluffy creature with an appearance reminiscent of a mythical being. The creature's fur texture is rendered in high detail. The monster's large eyes and open mouth express wonder and curiosity.

(f) Prompt: An isolated coloured abstract sculpture with a dali shape.

Figure 1: Text-(to-video)-to-4D samples generated by equipping Vidu4D with a pretrained video diffusion model [4]. For each sample, we exhibit per-frame 3D rendering for novel-view color, normal, and surfel feature. We observe that Vidu4D can reconstruct precisely detailed and photo-realistic 4D representation. See our accompanying videos in our project page for better visual quality.

However, achieving high-fidelity 4D reconstruction from generated videos poses great challenges. Non-rigidity and frame distortion are prevalent issues that can undermine the temporal and spatial coherence of the reconstructed content, thus complicating the creation of a seamless and coherent depiction of dynamic subjects.

In this work, we introduce Vidu4D, a novel reconstruction pipeline designed to accurately reconstruct 4D representations from single generated videos, facilitating the creation of 4D content with high precision in spatial and temporal coherence. Vidu4D contains two novel stages, namely, the initialization of non-rigid warping fields and Dynamic Gaussian Surfels (DGS), together enabling the reconstruction of high-fidelity 4D content with detailed appearance and accurate geometry.

Specifically, the proposed DGS optimizes non-rigid warping functions that transform Gaussian surfels from static to dynamically warped states. This dynamic transformation accurately represents motion and deformation over time, crucial for capturing realistic 4D representations. Besides, DGS demonstrates superior 4D reconstruction performance due to two other key aspects. Firstly, in terms of geometry, DGS adheres to Gaussian surfels principles [16, 28] to achieve precise geometric representation. Unlike existing methods, DGS incorporates warped-state normal consistency regularization to align surfels with actual surfaces with learnable continuous fields (*w.r.t.* spatial coordinate and time) to ensure smooth warping when estimating normals. Secondly, for appearance, DGS learns additional refinements on the rotation and scaling parameters of Gaussian surfels by a dual branch structure. This refinement reduces the flickering artifacts during warping and allows for the precise rendering of appearance details, resulting in high-quality reconstructed 4D representations.

By integrating Vidu4D with an existing powerful video generative model named Vidu [4], the overall framework demonstrates exceptional capabilities in text-to-4D generation. We provide 4D visualization results in Fig. 1. Extensive experiments based on the generated videos verify the effectiveness of our method compared to current state-of-the-art methods.

## 2 Related works

**3D representation.** Transforming 2D images into 3D representations has long been a central challenge in the field. Initially, triangle meshes were favored for their compactness and compatibility with rendering pipelines [9, 17, 67, 78, 82, 93]. However, the transition to more sophisticated volumetric methods was inevitable due to the limitations of surface-based approaches. Early volumetric representations included voxel grids [35, 48, 61, 72] and multi-plane images [20, 54, 74, 75, 80, 105], which, despite their straightforwardness, demanded intricate optimization strategies. The introduction of neural radiance fields (NeRF) [55] marked a great advancement, offering an implicit volumetric neural representation that could store and query the density and color of each point, leading to highly realistic reconstructions. The NeRF paradigm has since been improved upon in terms of reconstruction quality [5, 6, 33, 53, 92] and rendering [12, 23, 25, 27, 41, 45, 49, 64–66, 85, 102]. To address the limitations of NeRF, such as rendering speed and memory usage, recent work dubbed 3D Gaussian splatting (3DGS) [33] has proposed anisotropic Gaussian representations with GPU-optimized tile-based rasterization. This has opened up new avenues for surface extraction [24, 28], generation [14, 77, 96], and large-scale scene reconstruction [34, 46, 70], with 3DGS emerging as a universal representation for 3D scenes and objects. Gaussian surfels methods [16, 28] further exhibit advantages in modeling accurate geometry. While these methods have significantly advanced the field of static 3D representation, capturing the dynamic aspects of real-world scenes with non-rigid motion and deformation introduces a distinct set of challenges that demand innovative solutions.

**Dynamic reconstruction and generation.** The dynamic reconstruction of scenes from video captures presents a more complex challenge than static reconstruction, necessitating the capture of non-rigid motion and deformation over time [30, 37, 60, 76, 88]. Traditional methods have explored dynamic reconstruction using synchronized multi-view videos [1, 3, 11, 36, 48, 59, 73, 83, 84, 86, 89] or have focused on specific dynamic elements like humans or animals. More recently, there has been a shift towards reconstructing non-rigid objects from monocular videos, which is a more practical yet challenging scenario. One approach involves incorporating time as an additional input to the neural radiance field [11, 38, 68, 98], allowing for explicit querying of spatiotemporal information. Another line of research decomposes the spatiotemporal radiance field into a canonical space and a deformation field, representing spatial attributes and their temporal variations [18, 19, 19, 21, 22, 31, 39, 44, 47, 57, 58, 63, 69, 79, 95, 104]. With advancements in 3DGS, deformable-GS [100]

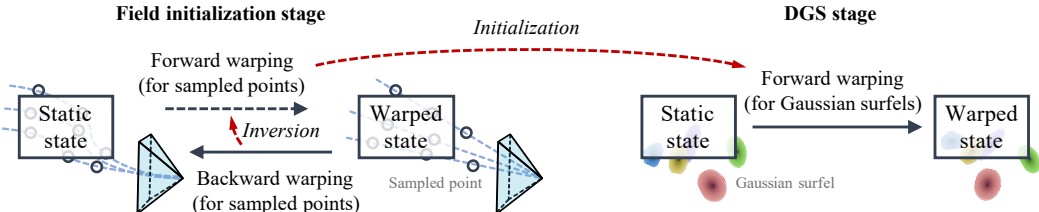

Figure 2: Illustration of the pipeline of Vidu4D, including the initialization stage and the DGS stage.

and 4DGS [94] have been developed, utilizing neural deformation fields with multi-layer perception (MLP) and triplane, respectively. SCGS [29] and dynamic 3D Gaussians [52] also advance the field by modeling time-varying scenes. Building on these advances, our work introduces dynamic Gaussian surfels, a novel extension of Gaussian representations that enhances the quality of both appearance and surface reconstruction under dynamic scenarios. A concurrent work DGM [43] builds time-consistent meshes from a monocular video with 3D Gaussian Splatting. In the realm of 3D or 4D generation, our approach diverges from recent progress in optimization-based [2, 13, 14, 40, 42, 62, 71, 88, 90], feed-forward [26, 91, 106], and multi-view reconstruction methods [15, 50, 51] by leveraging a video generative model to achieve generation capabilities. Our primary focus is on preserving high-quality appearance and geometrical integrity from generated videos. This results in a generation process that not only captures the nuances of motion and deformation but also maintains the high standards of realism and detail that are essential for creating immersive and lifelike virtual 3D representations.

## 3  Method

We start by introducing the problem definition for 4D reconstruction in Sec. 3.1. Following that, we introduce our Vidu4D which encompasses two novel stages. The first stage is designed to learn Dynamic Gaussian Surfels (DGS), ensuring precise representation of both visual appearance and geometric structure during the non-rigid reconstruction process, as detailed in Sec. 3.2. The second stage focuses on establishing the initial non-rigid warping fields of DGS, as detailed in Sec. 3.3.

### 3.1  Problem Definition

When given a single sequence of RGB video with $T$ frames, the goal of 4D reconstruction is to determine a sequential 3D representation that could be rendered to fit each video frame as much as possible. Specifically, suppose the 3D representation for the $t$-th frame (termed as time $t$) is parameterized by $\theta_t$, where $t = 1, \cdots, T$. Given a differentiable rendering mapping $\boldsymbol{g}$, we could obtain the rendered color at the frame pixel $\bar{\mathbf{x}}^t \in \mathbb{R}^2$. We choose volume rendering as commonly adopted in NeRF [55], Gaussian Splatting [33], and Gaussian Surfels [16, 28]. The optimization of 4D reconstruction can be implemented by minimizing the empirical loss as

$$\min_{\theta} \frac{1}{T} \sum_{t=1}^{T} \sum_{\bar{\mathbf{x}}^t} \mathcal{L}\Big(\mathbf{c}(\bar{\mathbf{x}}^t) = \boldsymbol{g}\big(\theta_t, \{\mathbf{x}_i^t\}_{i=1,\cdots,N}\big), \hat{\mathbf{c}}(\bar{\mathbf{x}}^t)\Big), \tag{1}$$

where $\mathbf{x}_i^t \in \mathbb{R}^3$ is the $i$-th 3D point sampled or intersected with Gaussian primitives along the ray that emanates from the frame pixel $\bar{\mathbf{x}}^t$; $N$ is the number of sampled or intersected points per ray; $\mathbf{c}(\bar{\mathbf{x}}^t)$ and $\hat{\mathbf{c}}(\bar{\mathbf{x}}^t)$ are the rendered color and the observed color at $\bar{\mathbf{x}}^t$, respectively.

In the following, we detail the proposed **Vidu4D**, a reconstruction pipeline comprising two key stages as illustrated in Fig. 2, including a field initialization stage and a DGS stage.

### 3.2  Dynamic Gaussian Surfels

By optimizing Eq. (1), essentially our goal is to build a sequential 3D representation that could deform to be consistent with each 2D frame. We first start by considering an ideal video exhibiting different views of the same static object without object deformation, movement, or video distortion. To model the 3D representation with high appearance fidelity and geometry accuracy, we follow the method of using differentiable 2D Gaussian primitives as proposed by recent Gaussian Surfels advances [16, 28].

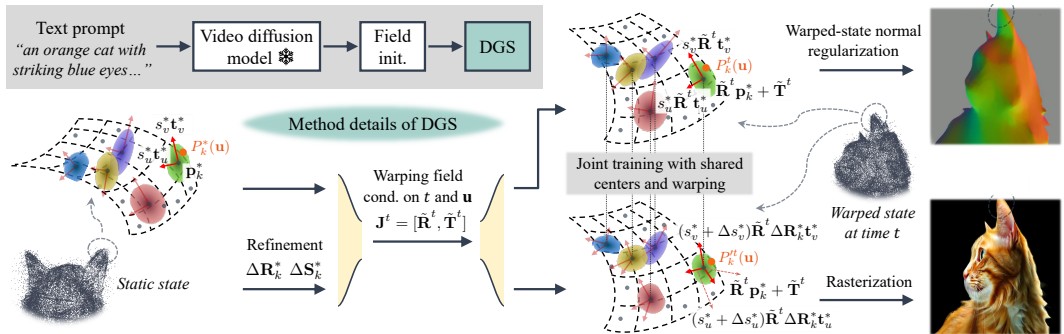

Figure 3: Illustration of the overall framework and our DGS in detail. For DGS, Gaussian surfels in the static state are transformed to the warped state by learning non-rigid warping functions conditioned on time $t$ and coordinate $\mathbf{u}$. We incorporate warped-state normal regularization for accurate geometry, and refined rotation and scaling matrices of Gaussian surfels for detailed appearance. Both branches in the warped state, including with and without refinement, share the same centers of Gaussian surfels and the same warping functions. "Field init." stands for field initialization as introduced in Sec. 3.3.

Specifically, the $k$-th Gaussian surfel (of the total $K$) is characterized by a central point $\mathbf{p}_k^* \in \mathbb{R}^3$ and a local coordinate system centered at $\mathbf{p}_k^*$ with two principal tangential vectors $\mathbf{t}_u^* \in \mathbb{R}^{3 \times 1}$, $\mathbf{t}_v^* \in \mathbb{R}^{3 \times 1}$ and scaling factors $s_u^* \in \mathbb{R}$, $s_v^* \in \mathbb{R}$. Here, we use the notation "$*$" to represent parameters in the static state. A Gaussian surfel is computed as a 2D Gaussian defined in a local tangent plane in the world space. Following [28], for any point $\mathbf{u} = (u, v)$ located on the $uv$ coordinate system centered at $\mathbf{p}_k^*$, its coordinate in the world space, denoted as $P_k^*(\mathbf{u}) \in \mathbb{R}^{3 \times 1}$, is computed by

$$P_k^*(\mathbf{u}) = \mathbf{p}_k^* + s_u^* \mathbf{t}_u^* u + s_v^* \mathbf{t}_v^* v = [\mathbf{R}_k^* \mathbf{S}_k^* \quad \mathbf{p}_k^*] (u, v, 1, 1)^\top, \tag{2}$$

where $\mathbf{R}_k^* = [\mathbf{t}_u^*, \mathbf{t}_v^*, \mathbf{t}_u^* \times \mathbf{t}_v^*] \in SO(3)$ denotes the rotation matrix, and the diagonal matrix $\mathbf{S}_k^* = \mathrm{diag}(s_u^*, s_v^*, 0) \in \mathbb{R}^{3 \times 3}$ denotes the scaling matrix.

In this work, our focus is on 4D reconstruction from a single generated video, which may exhibit large non-rigidity, distortion, or illumination changes. We introduce **Dynamic Gaussian Surfels (DGS)**, a method designed to achieve precise 4D reconstruction while accommodating non-rigidity and other time-varying effects.

Motivated by recent advancements in non-rigid reconstruction methods [57, 88, 98], we aim to ensure that the target object maintains a consistent static state across different frames, thereby mitigating non-rigidity and distortion effects. To achieve this, we employ warping techniques on each Gaussian surfel represented by $P_k^*(\mathbf{u})$, transforming them into a corresponding Gaussian surfel $P_k^t(\mathbf{u})$ at time $t$, which is centered at $\mathbf{p}_k^t \in \mathbb{R}^3$ with a rotation matrix $\mathbf{R}_k^t \in SO(3)$ and a scaling matrix $\mathbf{S}_k^t \in \mathbb{R}^{3 \times 3}$.

**Non-rigid warping for Gaussian surfels.** We now build the warping process from the static state to the warped state. We leverage a non-rigid warping function with $B$ bones as key points to ease the training of deformation. In the static state, the $b$-th bone is represented by 3D Gaussian ellipsoids [97], with more details provided in the Appendix. We let $\mathbf{J}_b^t \in SE(3)$ represent a rigid transformation that moves the $b$-th bone from its static state to the warped state at time $t$. In effect, $\mathbf{J}_b^t$ is achieved by non-linear mappings using a multi-layer perception (MLP) with $SE(3)$ guaranteed, as will be given later in Eq. (5). The non-rigid warping function can be written as the weighted combination of $\mathbf{J}_b^t$, where we apply dual quaternion blend skinning (DQB) [32] to ensure valid $SE(3)$ after combination,

$$\mathbf{J}^t = \mathcal{R}\Big(\sum_{b=1}^{B} w_b^t \mathcal{Q}(\mathbf{J}_b^t)\Big), \tag{3}$$

where $w_b^t$ is the $b$-th element of the skinning weight vector $\mathbf{w}^t \in \mathbb{R}^{B \times 1}$, as detailed in the Appendix; $\mathcal{Q}$ and $\mathcal{R}$ denote the quaternion process and the inverse quaternion process, respectively. In this case, there is $\mathbf{J}^t \in SE(3)$.

We therefore rewrite the warping as $\mathbf{J}^t = [\tilde{\mathbf{R}}^t, \tilde{\mathbf{T}}^t]$ with the rotation $\tilde{\mathbf{R}}^t \in SO(3)$ and translation $\tilde{\mathbf{T}}^t \in \mathbb{R}^3$, and apply the corresponding transformation to Eq. (2) by

$$P_k^t(\mathbf{u}) = \mathbf{J}^t P_k^*(\mathbf{u}) = [\tilde{\mathbf{R}}^t \mathbf{R}_k^* \mathbf{S}_k^* \quad \tilde{\mathbf{R}}^t \mathbf{p}_k^* + \tilde{\mathbf{T}}^t] (u, v, 1, 1)^\top. \tag{4}$$

Note that Eq. (4) holds for any given point $P_k^*(\mathbf{u})$ including the center point of the $k$-th Gaussian surfel (*i.e.*, $\mathbf{p}_k^*$) when $\mathbf{u} = (0,0)$. By deriving Eq. (4), we enable connection of the warping function *w.r.t.* to any point $\mathbf{u} = (u,v)$ on the local coordinate system centered at $\mathbf{p}_k^*$, which is needed later in Eq. (8) where $\mathbf{u}$ is an intersection with Gaussian surfels and a ray that emanates from the frame pixel.

**Warped-state normal regularization.** To accurately capture the geometric representation, we follow similar methods in Gaussian Surfels [16, 28] to add normal consistency regularization which encourages all Gaussian surfels to be locally aligned with the actual surfaces. Differently, unlike 3D reconstruction for static scenes, 4D reconstruction commonly faces non-rigidity and distortion. Thus simply performing regularization to promote surface-aligned Gaussian surfels like previous methods harms the structural integrity due to the non-rigid warping.

We therefore design a warped-state normal regularization. As mentioned, each point $P_k^t(\mathbf{u})$ in the warped state at time $t$ is transformed from its corresponding static point $P_k^*(\mathbf{u})$ based on the warping function in Eq. (4), namely, $P_k^t(\mathbf{u}) = \mathbf{J}^t P_k^*(\mathbf{u})$ with $\mathbf{J}^t$ composed by $\mathbf{J}_b^t$. To maintain the structural integrity to a large extent when regularizing normal, we design $\mathbf{J}_b^t$ as a continuous field that takes both the point $P_k^*(\mathbf{u})$ (or equivalently, $\mathbf{u}$ in the local coordinate system) and the time $t$ as conditions. By this setting, $\mathbf{J}_b^t$ is expected to change continuously with the change of $\mathbf{u}$ or $t$. We implement the continuous field by using a NeRF-style MLP which directly outputs a 6-dimensional dual quaternion, and rely on the inverse quaternion process $\mathcal{R}$ to guarantee SE(3), *i.e.*,

$$\mathbf{J}_b^t = \mathcal{R}\big(\mathbf{MLP}(\boldsymbol{\gamma}_b^t; \mathbf{u}, t)\big), \tag{5}$$

where $\boldsymbol{\gamma}_b^t$ is a learnable latent code for encoding the $b$-th bone at time $t$; both $\mathbf{u}$ and $t$ are sent to the MLP as conditions to obtain $\mathbf{J}_b^t$. Thus $\mathbf{J}^t$ is also expected to be continuous *w.r.t.* $\mathbf{u}$ and $t$.

Based on the above design, the normal consistency loss at time $t$ is obtained similar to [28],

$$\mathcal{L}_n = \sum_k \omega_k (1 - \mathbf{n}_k^\top \mathbf{N}^t), \quad \mathbf{N}^t(x, y) = \frac{\nabla_x \mathbf{p}^t \times \nabla_y \mathbf{p}^t}{|\nabla_x \mathbf{p}^t \times \nabla_y \mathbf{p}^t|}, \tag{6}$$

where $k$ indexes over intersected surfels along the ray that emanates from the frame pixel $\bar{\mathbf{x}}$; $\omega_k = \alpha_k \, \mathcal{G}_k(\mathbf{u}(\bar{\mathbf{x}})) \prod_{j=1}^{k-1}(1 - \alpha_j \, \mathcal{G}_j(\mathbf{u}(\bar{\mathbf{x}})))$ denotes the blending weight of the intersection point; $\mathbf{n}_k$ represents the normal of the surfel that is oriented towards the camera; $\mathbf{N}^t$, computed with finite differences, is the surface normal estimated by the nearby depth point $\mathbf{p}^t$ at warped state time $t$.

In summary, by learning a continuous warping field and aligning the surfel normal with the estimated surface normal in the warped state, we ensure that all Gaussian surfels locally approximate the actual object surface without being noticeably impaired by the non-rigid warping.

**Dual branch structure with refinement.** To further achieve fine-grained appearance and reduce the texture flickering during warping, we propose to learn refinement terms for adjusting the rotation matrices $\mathbf{R}_k^*$ and scaling matrices $\mathbf{S}_k^*$ (defined in Eq. (2)) in the static state. We suppose the refinement terms are $\Delta\mathbf{R}_k^* \in \mathrm{SO}(3)$ and $\Delta\mathbf{S}_k^* \in \mathbb{R}^{3\times3}$, respectively. Note that the third-axis of $\Delta\mathbf{S}_k^*$ is no longer necessarily 0. During refinement, we remain the center points $\mathbf{p}_k^*$ and the warping $\mathbf{J}^t$ (*i.e.*, including both $\tilde{\mathbf{R}}^t$ and $\tilde{\mathbf{T}}^t$) to be unchanged. The new warped process is formulated as,

$$P_k'^t(\mathbf{u}) = \big[\tilde{\mathbf{R}}^t(\Delta\mathbf{R}_k^*\mathbf{R}_k^*)(\mathbf{S}_k^* + \Delta\mathbf{S}_k^*) \quad \tilde{\mathbf{R}}^t\mathbf{p}_k^* + \tilde{\mathbf{T}}^t\big](u, v, 1, 1)^\top. \tag{7}$$

During the training of DGS, we maintain two branches including one with refinement and one without. In the warped state, both branches are jointly trained with shared warping functions and centers of Gaussian primitives[2]. Due to the involvement of $\Delta\mathbf{R}_k^*$ and $\Delta\mathbf{S}_k^*$, both branches have different rotation and scaling matrices of Gaussian primitives.

**Rasterization.** Given a frame pixel $\bar{\mathbf{x}}$ and a camera ray that emanates from $\bar{\mathbf{x}}$, following the static-state methods to calculate intersection coordinates with Gaussian primitives along the ray [28, 33], we could obtain warped-state intersection coordinates based on Eq. (4) and Eq. (7). We then perform the volume rendering process [28] that integrates alpha-weighted appearance along the ray by

$$\mathbf{c}(\bar{\mathbf{x}}) = \sum_k \mathbf{c}_k \, \alpha_k \, \mathcal{G}_k\big(\mathbf{u}(\bar{\mathbf{x}})\big) \prod_{j=1}^{k-1} \big(1 - \alpha_j \, \mathcal{G}_j\big(\mathbf{u}(\bar{\mathbf{x}})\big)\big), \tag{8}$$

---

[2]Here, since the third-axis of the refined scaling matrix is not necessarily 0, we adopt "Gaussian primitive" for commonly referring to both Gaussian surfel and the refined Gaussian.

where $k$ indexes over intersected Gaussian primitives along the ray that emanates from the frame pixel $\bar{\mathbf{x}}$; $\alpha_k$ and $\mathbf{c}_k$ denote the opacity and view-dependent appearance parameterized with spherical harmonics of the $k$-th Gaussian surfel, respectively; $\mathcal{G}_k(\mathbf{u}(\bar{\mathbf{x}})) = \exp\left(-\frac{u^2+v^2}{2}\right)$ corresponds to the $k$-th intersection point $\mathbf{u}(\bar{\mathbf{x}})$ which could be directly calculated when given $P_k^t(\mathbf{u})$ or $P_k'^t(\mathbf{u})$ and the corresponding local coordinate system. During implementation, $\mathcal{G}_k(\mathbf{u}(\bar{\mathbf{x}}))$ is further applied a low-pass filter following [7, 28].

A detailed architecture of DGS is depicted in Fig. 3. Important symbols are summarized in our Appendix.

### 3.3   Field Initialization

Given that the camera trajectory of generated videos is unknown, SfM methods like COLMAP struggle to converge due to rigidity violations. Additionally, since the background of generated videos appears to exhibit soft deformation or flickering colors, proper estimation of camera/body poses through background SfM is hindered. These challenges often result in very few successful registrations, as demonstrated in previous monocular 4D reconstruction tasks [98].

To address this, we design an implicit field before performing DGS to initialize the camera poses and establish the continuous warping field in Eq. (5). In this part, we propose the **field initialization** as another key component of our pipeline to initialize the continuous warping field of DGS for fast and stable convergence, as detailed below.

Initially, we train a neural Signed Distance Function (SDF) model [87], leveraging the same warping structure with bones as utilized in DGS. While DGS transforms Gaussian surfels from the static state to the warped state for rasterization, the neural SDF reverses this process, mapping points along camera rays from the warped state back to the static state. For the neural SDF component, we optimize the reverse warping process and deduce the forward warping as its inverse by minimizing a cycle loss, inspired by [10, 98]. Subsequently, we initialize $\mathbf{MLP}(\cdot)$ in Eq. (5) to obtain warping functions $\mathbf{J}_b^f$ by the network weights learned by the neural SDF part.

During the rendering of the neural SDF, we perform backward warping on the warped-state sampling points to the static state,

$$\mathbf{J}^{f,-1} = \mathcal{R}\Big(\sum_{b=1}^{B} w_b^f \mathcal{Q}(\mathbf{J}_b^f)^{-1}\Big), \quad \mathbf{X}^f = \mathbf{J}^{f,-1}\mathbf{X}^*, \tag{9}$$

which is an inversion of Eq. (3). By querying the SDF with a sample point $\mathbf{X}^*$ in the static state, we render RGB and compute the photometric loss to optimize the SDF and the warping field defined in Eq. (5).

Nevertheless, there are two discrepancies between the neural SDF warping and DGS warping. Firstly, sampling points of the neural SDF are distributed in the frustum of the camera, while sampling points of DGS are distributed on the object surface. Additionally, we train the inversion of during initialization, while we utilize the non-inverse ones in DGS. To resolve the distribution gap and ensure that faithfully models the forward warping, we add a cycle loss,

$$\mathcal{L}_{\mathrm{cyc}} = \big\|\mathbf{J}^f(\mathbf{J}^{f,-1}(\mathbf{X}^f)) - \mathbf{X}^f\big\|_2^2, \tag{10}$$

where $\mathbf{X}^f$ could be either a mesh surface point or a sample point on the camera ray.

After initialization, we extract the canonical space mesh using marching cubes and initialize Gaussian surfels on it. We set the spherical harmonic in 0-th order to the RGB value of the nearest vertices. The warping field and learned camera poses are retained.

With the field initialization before DGS, our Vidu4D is capable of performing a text-(to-video)-to-4D generation task with the integration of existing video diffusion models.

## 4   Experiment

In this section, we provide an extensive evaluation of our method DGS (Sec. 3.2) with the initialization in Sec. 3.3, comparing both appearance and geometry against previous state-of-the-art methods. Additionally, we analyze the contributions of each proposed component in detail.

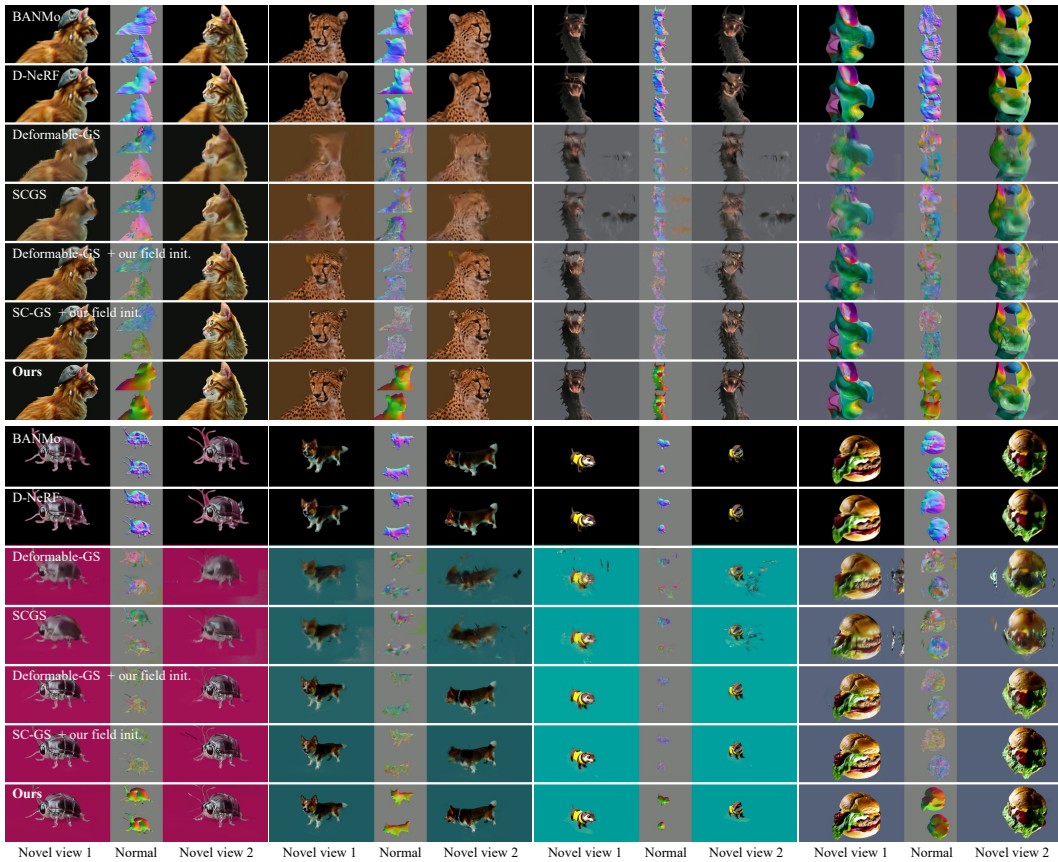

| Novel view 1 | Normal | Novel view 2 | Novel view 1 | Normal | Novel view 2 | Novel view 1 | Normal | Novel view 2 | Novel view 1 | Normal | Novel view 2 |
|---|---|---|---|---|---|---|---|---|---|---|---|

Figure 4: Novel-view qualitative evaluation compared with SOTA methods including NeRF-based methods (BANMo [98] and D-NeRF [63]) and Gaussian splatting-based methods (Deformable-GS [100] and SCGS [29]). We also provide our learned camera poses to baseline approaches for a fair comparison. These variants are denoted as "w. Poses". Best view in color and zoom in.

## 4.1 Implementation

For all qualitative and quantitative experiments, we follow the standard pipeline for dynamic reconstruction [58], to construct our evaluation setup by selecting every fourth frame as a training frame and designating the middle frame between each pair of training frames as a validation frame.

Our model configuration involves several key parameters to balance reconstruction and regularization losses. For the field initialization stage, we use a similar architecture with 8 layers for volume rendering as in NeRF [55], and initialize MLP for predicting SDF as an approximate unit sphere [101]. We obtain a neural SDF, a warping field, and camera poses after this stage. For the DGS stage, we initialize centers of the Gaussian surfels with the sampled surface points extracted from the neural SDF, and initialize the warping field by the forward field from the first stage. The dimension of the latent code embedding $\gamma_b^t$ is set as 128. Following BANMo [98], we adopt 25 bones to optimize skinning weights. For each reconstruction, the overall training takes over 1 hour on an A800 GPU.

## 4.2 Qualitative Evaluation

In the qualitative evaluation, we visually compare the novel-view reconstructions produced by our DGS against those generated by other state-of-the-art models, as illustrated in Fig. 4. Our evaluation focuses on several key aspects including detail preservation, texture quality, and geometric accuracy. Compared to methods based on implicit fields, the integration of Gaussian in our approach facilitates the rendering of highly detailed textures. Additionally, benefiting from a more geometry-aware representation, our method produces superior normal maps compared to those purely Gaussian-based methods. This also enhances the robustness of our method against artifacts of the generated videos

Table 1: Novel-view quantitative results on generated videos. Evaluation metrics are PSNR, SSIM, and LPIPS. We report results on three single videos and the averaged results over 30 single videos.

| | Cat | | | Cheetah | | | Dragon | | | Average over 30 videos | | |
|---|---|---|---|---|---|---|---|---|---|---|---|---|
| | PSNR ↑ | SSIM ↑ | LPIPS ↓ | PSNR ↑ | SSIM ↑ | LPIPS ↓ | PSNR ↑ | SSIM ↑ | LPIPS ↓ | PSNR ↑ | SSIM ↑ | LPIPS ↓ |
| BANMo [98] | 15.10 | 0.6514 | 0.2575 | 13.15 | 0.5921 | 0.3241 | 18.48 | 0.6423 | 0.3500 | $13.62 \pm 2.99$ | $0.6153 \pm 0.0714$ | $0.3738 \pm 0.0665$ |
| D-NeRF [63] | 15.15 | 0.6537 | 0.2657 | 13.21 | 0.5930 | 0.3344 | 18.53 | 0.6489 | 0.3527 | $21.01 \pm 2.86$ | $0.8519 \pm 0.0717$ | $0.1522 \pm 0.0754$ |
| Deformable-GS [100] | 19.09 | 0.7815 | 0.2434 | 20.35 | 0.8039 | 0.1982 | 24.19 | 0.9100 | 0.0992 | $13.22 \pm 3.42$ | $0.5934 \pm 0.0535$ | $0.3749 \pm 0.0763$ |
| SCGS [29] | 19.46 | 0.7867 | 0.2405 | 20.87 | 0.8123 | 0.1919 | 24.03 | 0.9083 | 0.1009 | $21.17 \pm 2.69$ | $0.8547 \pm 0.0691$ | $0.1504 \pm 0.0737$ |
| Deformable-GS + our field init. | 21.94 | 0.8123 | 0.1816 | 22.41 | 0.8200 | 0.1687 | 26.05 | 0.9218 | 0.0894 | $22.63 \pm 2.14$ | $0.8469 \pm 0.0438$ | $0.1452 \pm 0.0354$ |
| SCGS + our field init. | 23.25 | 0.8268 | 0.1574 | 23.70 | 0.8338 | 0.1497 | 28.40 | 0.9375 | 0.0686 | $24.75 \pm 2.11$ | $0.8680 \pm 0.0440$ | $0.1201 \pm 0.0359$ |
| **Ours** | **24.63** | **0.8432** | **0.1559** | **25.68** | **0.8843** | **0.1117** | **28.58** | **0.9392** | **0.0618** | $\mathbf{27.30 \pm 2.66}$ | $\mathbf{0.9152 \pm 0.0602}$ | $\mathbf{0.0877 \pm 0.0564}$ |

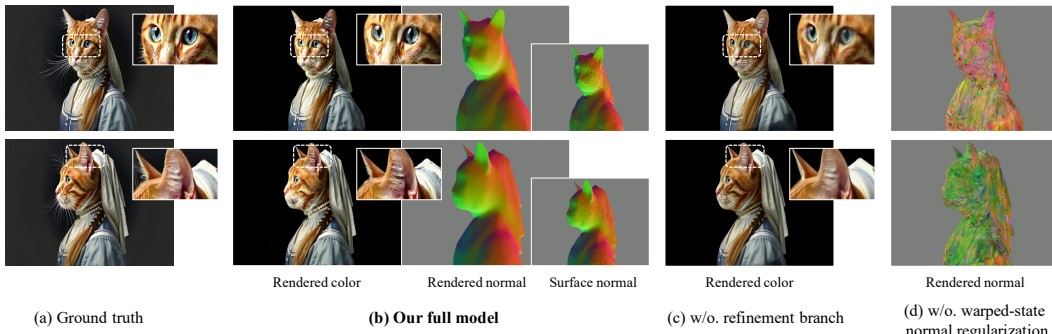

(a) Ground truth     Rendered color    Rendered normal    Surface normal     Rendered color     Rendered normal

(a) Ground truth     (b) **Our full model**     (c) w/o. refinement branch     (d) w/o. warped-state normal regularization

Figure 5: Ablation studies on the geometric regularization and refinement strategy. For our full model shown in (b), we provide our rendered color, rendered normal, and surface normal (estimated from the depth points for regularization). Additionally, for comparison, we visualize the rendered color for the case without refinements in (c) and the rendered normal for the case without warped-state normal regularization in (d), respectively. We showcase our model's fidelity with close-ups.

like occlusions. For instance, in the third clip of the series, which features a dragon shrouded in fog, both SCGS and Deformable-GS methods tend to overfit and subsequently show a decline in performance. In contrast, our method consistently delivers superior results.

### 4.3 Quantitative Evaluation

We provide the quantitative evaluation comparing our method with state-of-the-art works in Table 1. Metrics include Peak Signal-to-Noise Ratio (PSNR) to evaluate the fidelity of the reconstructed textures, Structural Similarity Index (SSIM) for the quality evaluation, and LPIPS [103] as a perceptual metric. Our method exhibits superiority over all baseline methods, even with our learned poses, *e.g.*, ∼2.5 PSNR increase over SCGS with poses for the averaged results.

### 4.4 Ablations

To understand the contributions of each component in Vidu4D, especially DGS, we conduct ablation studies in this section. We remove or alter specific elements of our model and observe the resulting performance changes in both appearance and geometry reconstruction.

**Geometric regularization.** We evaluate the impact of warped-state normal regularization by disabling it during training. From Fig. 5(b)(d), we observe that when removing the regularization, there is an obvious degradation in the structural integrity of surface-aligned Gaussian surfels, leading to noticeable inconsistency in the reconstructed 4D models.

**Refinement strategy.** We examine the effect of omitting refinements by keeping one branch (the concept of branches could be better visualized in Fig. 3) during training, shown in Fig. 5(b)(c). The performance indicates that removing refinements increases the loss of fine-grained appearance details. Additionally, we also find that refinements are crucial for mitigating the texture flickering issue.

**Additional ablations.** Please refer to the Appendix for additional ablation studies that detail the effectiveness of our refinement strategy and field initialization.

# 5 Conclusion

We introduce Vidu4D as a novel reconstruction model to achieve high-fidelity 4D representations from single generated videos. Vidu4D is powerful with our proposed DGS which builds the non-rigid warping field to transform Gaussian surfels, ensuring precise capture of motion and deformation over time. DGS also introduces key innovations that greatly enhance the accuracy and fidelity of 4D reconstruction, including dual branch refinement and warped-state geometric regularization. Our experiments demonstrate that Vidu4D outperforms existing methods in both quantitative and qualitative evaluations, highlighting its superiority in generating realistic and immersive 4D content.

**Limitations and broader impact.** While Vidu4D with DGS presents a significant performance in 4D reconstruction, currently there are still limitations such as the reliance on video quality, scalability challenges for large scenes, and computational difficulties in real-time applications. Additionally, when equipping Vidu4D with generative models, as with any generative technology, there is a risk of producing deceptive content which needs more caution.

# 6 Acknowledgement

This work was partly supported by the NSFC Projects (Nos. 62350080, 62306163, 62276149, 92370124, 92248303, U2341228, 62061136001, 62076147), BNRist (BNR2022RC01006), Tsinghua Institute for Guo Qiang, and the High Performance Computing Center, Tsinghua University. J. Zhu was also supported by the XPlorer Prize. Y.K. Wang was also supported by the China National Postdoctoral Program (No. 2023M741951) and Major Project of the New Generation of Artificial Intelligence (No. 2018AAA0102900).

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

# A    Appendix / Supplemental Material

Table 2: A summary of important symbols in DGS.

| Symbol | Definition and Usage |
|---|---|
| $\mathbf{t}_u^* \in \mathbb{R}^{3\times1}, \mathbf{t}_v^* \in \mathbb{R}^{3\times1}$ | Principal tangential vectors in the static state. |
| $s_u^* \in \mathbb{R}, s_v^* \in \mathbb{R}$ | Scaling factors in the static state. |
| $\mathbf{p}_k^* \in \mathbb{R}^{3\times1}$ | Center point coordinate (world space) of the $k$-th Gaussian surfel in the static state. |
| $P_k^*(\mathbf{u}) \in \mathbb{R}^{3\times1}$ | Coordinate (world space) in the static state, given $\mathbf{u} = (u,v)$ on the local $uv$ coordinate system centered at $\mathbf{p}_k^*$. |
| $\mathbf{R}_k^* = [\mathbf{t}_u^*, \mathbf{t}_v^*, \mathbf{t}_u^* \times \mathbf{t}_v^*] \in \mathrm{SO}(3)$ | Rotation matrix of the $k$-th Gaussian surfel in the static state. |
| $\mathbf{S}_k^* = \mathrm{diag}(s_u^*, s_v^*, 0) \in \mathbb{R}^{3\times3}$ | Scaling matrix of the $k$-th Gaussian surfel in the static state, a diagonal matrix. |
| $\mathbf{p}_k^t \in \mathbb{R}^{3\times1}$ | Center point coordinate (world space) of the $k$-th Gaussian surfel in the warped state. |
| $P_k^t(\mathbf{u}) \in \mathbb{R}^{3\times1}$ | Coordinate (world space) in the warped state, given $\mathbf{u} = (u,v)$ on the local $uv$ coordinate system centered at $\mathbf{p}_k^t$. |
| $\mathbf{c}_b^* \in \mathbb{R}^{3\times1}, \mathbf{V}_b^* \in \mathbb{R}^{3\times3}, \mathbf{\Lambda}_b^* \in \mathbb{R}^{3\times3}$ | Center, rotation matrix, and diagonal scaling matrix of the $b$-th Gaussian ellipsoid bone. |
| $\mathbf{w}^t \in \mathbb{R}^{B\times1}$ | Skinning weight vectors. |
| $\mathbf{J}_b^t \in \mathrm{SE}(3)$ | A rigid transformation that moves the $b$-th bone from its static state to the warped state at time $t$. |
| $\mathbf{J}^t = [\bar{\mathbf{R}}^t, \bar{\mathbf{T}}^t] \in \mathrm{SE}(3)$ | The warping function, a weighted combination of $\mathbf{J}_b^t$. |
| $\mathcal{Q}, \mathcal{R}$ | The quaternion process and the inverse quaternion process. |
| $\boldsymbol{\omega}_b^t \in \mathbb{R}^{128}$ | A learnable latent code for representing the body pose at time $t$. |
| $\mathbf{n}_k \in \mathbb{R}^{3\times1}$ | The normal of the $k$-intersected Gaussian surfel that is oriented towards the camera. |
| $\mathbf{N}^t \in \mathbb{R}^{3\times1}$ | The surface normal estimated by the nearby depth point $\mathbf{p}^t$ at warped state time $t$. |
| $\Delta\mathbf{R}_k^* \in \mathrm{SO}(3)$ | Learnable refinement term for adjusting $\mathbf{R}_k^*$. |
| $\Delta\mathbf{S}_k^* \in \mathrm{SO}(3)$ | Learnable refinement term for adjusting $\mathbf{S}_k^*$. |

## A.1    Details of Skinning Representation

As mentioned in the main paper, the warping process from the static state to the warped state is modelled as a time-varying non-rigid warping function with $B$ bones to be key points. In the static state, the $b$-th bone is represented by 3D Gaussian ellipsoids [97] with the center $\mathbf{c}_b^* \in \mathbb{R}^{3\times1}$, rotation matrix $\mathbf{V}_b^* \in \mathbb{R}^{3\times3}$, and diagonal scaling matrix $\mathbf{\Lambda}_b^* \in \mathbb{R}^{3\times3}$. For a 3D point $P_k^*(\mathbf{u})$, the skinning weight vectors $\mathbf{w}^t \in \mathbb{R}^{B\times1}$ at time $t$ is calculated by the normalized Mahalanobis distance following [98]

$$m_b^t = \left(P_k^*(\mathbf{u}) - \mathbf{c}_b^t\right)^\top \mathbf{Q}_b^t \left(P_k^*(\mathbf{u}) - \mathbf{c}_b^t\right), \quad \mathbf{w}^t = \sigma_{\mathrm{softmax}}\left(m_1^t, m_2^t, \cdots, m_B^t\right)^\top, \qquad (11)$$

where $m_b^t$ denotes the squared distance between $P_k^*(\mathbf{u})$ and the $b$-th bone; $\mathbf{c}_b^t \in \mathbb{R}^{3\times1}$ is the center of the $b$-th bone at time $t$, and $\mathbf{Q}_b^t = \mathbf{V}_b^{t\top}\mathbf{\Lambda}_b^*\mathbf{V}_b^t$ is the precision matrix composed by the bone orientation matrix $\mathbf{V}_b^t \in \mathbb{R}^{3\times3}$ at time $t$ and $\mathbf{\Lambda}_b^*$. Specifically, there is $(\mathbf{V}_b^t|\mathbf{c}^t) = \mathbf{J}_b^t(\mathbf{V}_b^*|\mathbf{c}^*)$ with $\mathbf{c}_b^*$, $\mathbf{V}_b^*$, and $\mathbf{\Lambda}_b^*$ being learnable parameters. $\sigma_{\mathrm{softmax}}$ is the softmax function.

## A.2    Ablation Studies of Field Initialization and Refinement

In dynamic videos captured in the wild, one of the primary challenges is the initialization of camera poses. In synthetic videos, preserving temporal consistency in texture and geometry is problematic, which significantly complicates the task of camera registration. To address this, we utilize an implicit field to both initialize the camera poses and establish the warping field. Initially, we estimate the transformation for each frame, followed by the computation of coarse camera poses through an iterative process. Subsequently, we adopt the approach outlined in NeuS [87] for scene representation. Feature extraction is performed using DinoV2 [56], facilitating unsupervised registration. To enhance this process, we train an additional channel in NeuS specifically for rendering features, which are then employed for registration purposes as described in RAC [99]. The camera poses without initialization and refined camera poses are depicted in Fig. 6. Without field initialization, the performance of DGS will degrade, as shown in Table 3. Also, please refer to the quantitative ablation of refinement in Table 3.

## A.3    Additional Qualitative Comparison

In this section, we present a detailed comparison of our results with previous works, as illustrated in Fig. 7-10. Our method consistently achieves high-quality texture details while maintaining smooth and realistic geometry.

Table 3: Quantitative ablation studies of the initialization and refinement.

| | Cat | | | Cheetah | | | Dragon | | |
|---|---|---|---|---|---|---|---|---|---|
| | PSNR ↑ | SSIM ↑ | LPIPS ↓ | PSNR ↑ | SSIM ↑ | LPIPS ↓ | PSNR ↑ | SSIM ↑ | LPIPS ↓ |
| Ours w.o. init. | 20.15 | 0.7961 | 0.2393 | 20.96 | 0.8194 | 0.1940 | 25.33 | 0.9146 | 0.0938 |
| Ours w.o. refinement | 24.19 | 0.8196 | 0.1797 | 24.10 | 0.8582 | 0.1242 | 27.71 | 0.9128 | 0.0687 |
| **Ours full** | **24.63** | **0.8432** | **0.1559** | **25.68** | **0.8843** | **0.1117** | **28.58** | **0.9392** | **0.0618** |

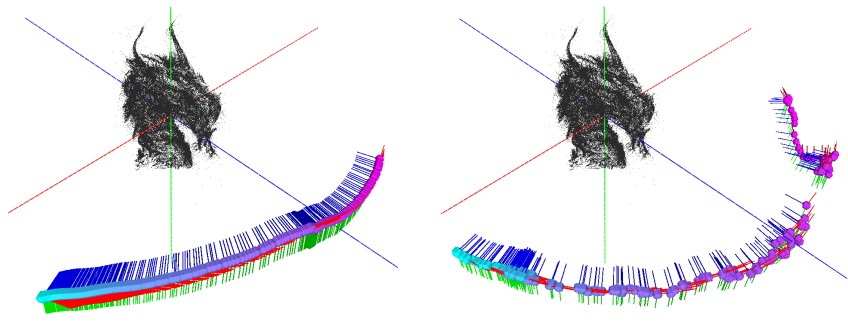

(a) Camera poses without field initialization        (b) Refined camera poses

Figure 6: Coarse camera poses and refined camera poses.

## A.4 Interpolation on Time and Views

We present results for interpolation on time and views, as illustrated in Fig. 11 and Fig. 12.

## A.5 Broader Impact

Generative models used in video generation might pose risks, for example, the potential for creating deepfakes or other misleading content that could be used for harmful purposes like misinformation, privacy invasion, or defamation. To mitigate these risks, we have chosen to release only the reconstruction code, deliberately avoiding the release of components that could facilitate the generation of content with ethical concerns. This decision ensures that our contribution is focused on advancing reconstruction techniques without enabling the creation of new, potentially harmful video content.

Besides, we have carefully considered the potential ethical risks associated with generative models, particularly in video content creation. To address these concerns, our model includes robust safety mechanisms designed to screen and prevent any misuse. We believe these measures effectively mitigate potential ethical risks and align with the standards of the community.

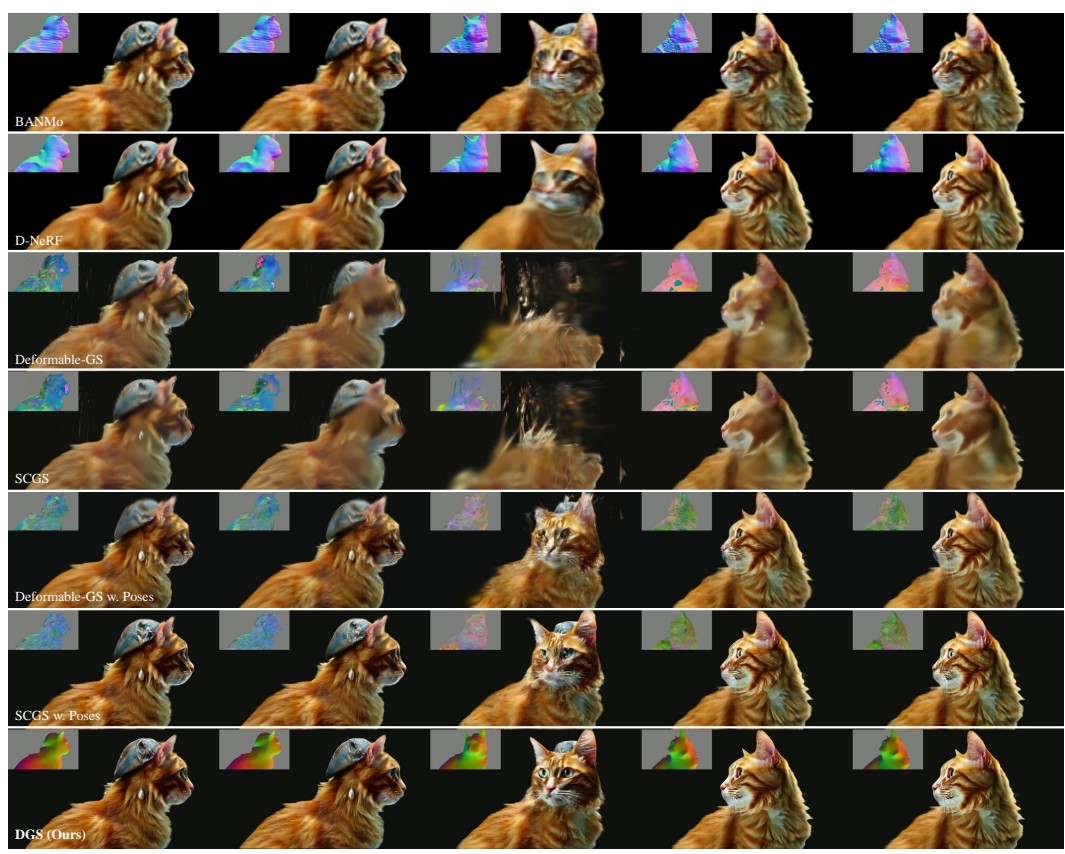

Figure 7: Additional qualitative comparison with more novel views.

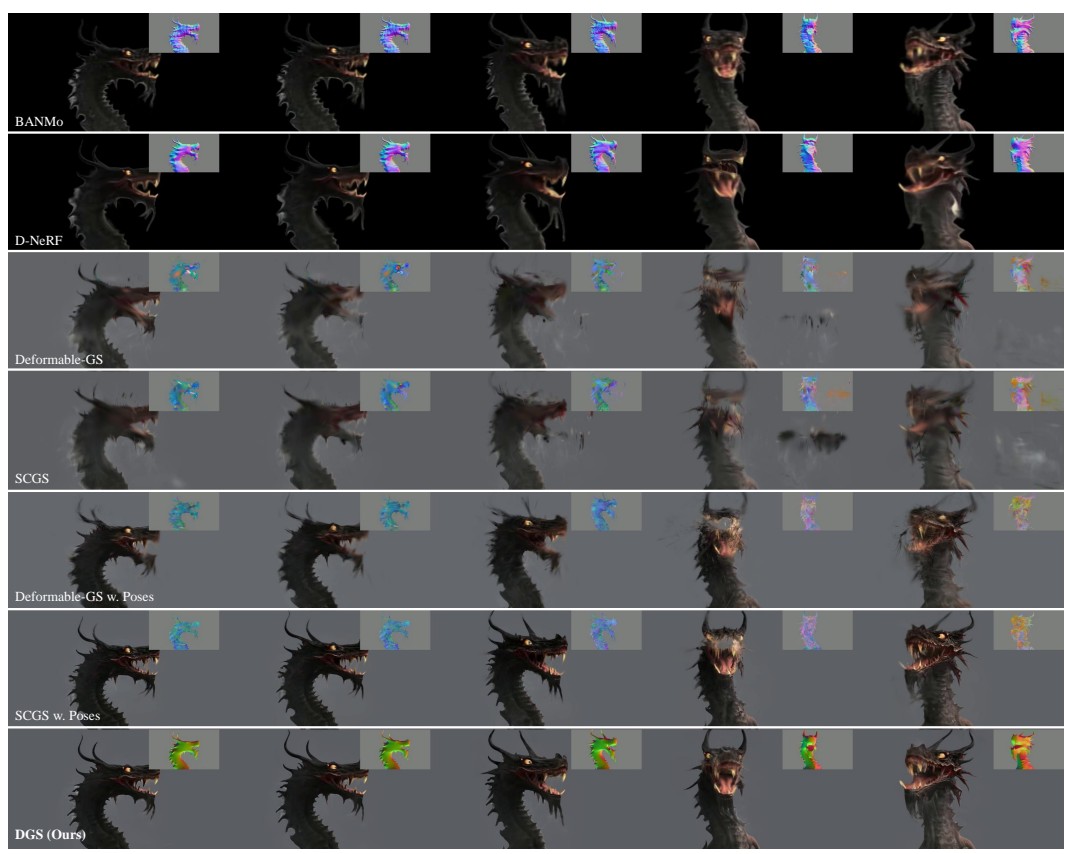

Figure 8: Additional qualitative comparison with more novel views.

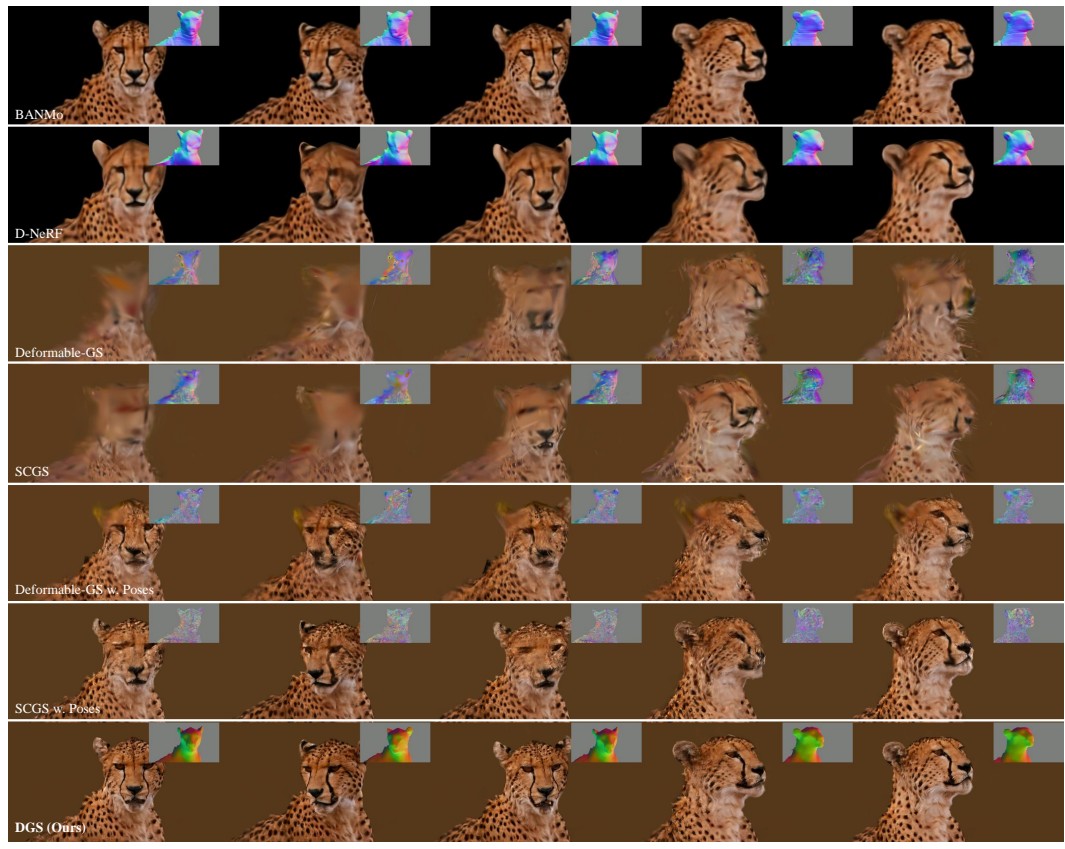

Figure 9: Additional qualitative comparison with more novel views.

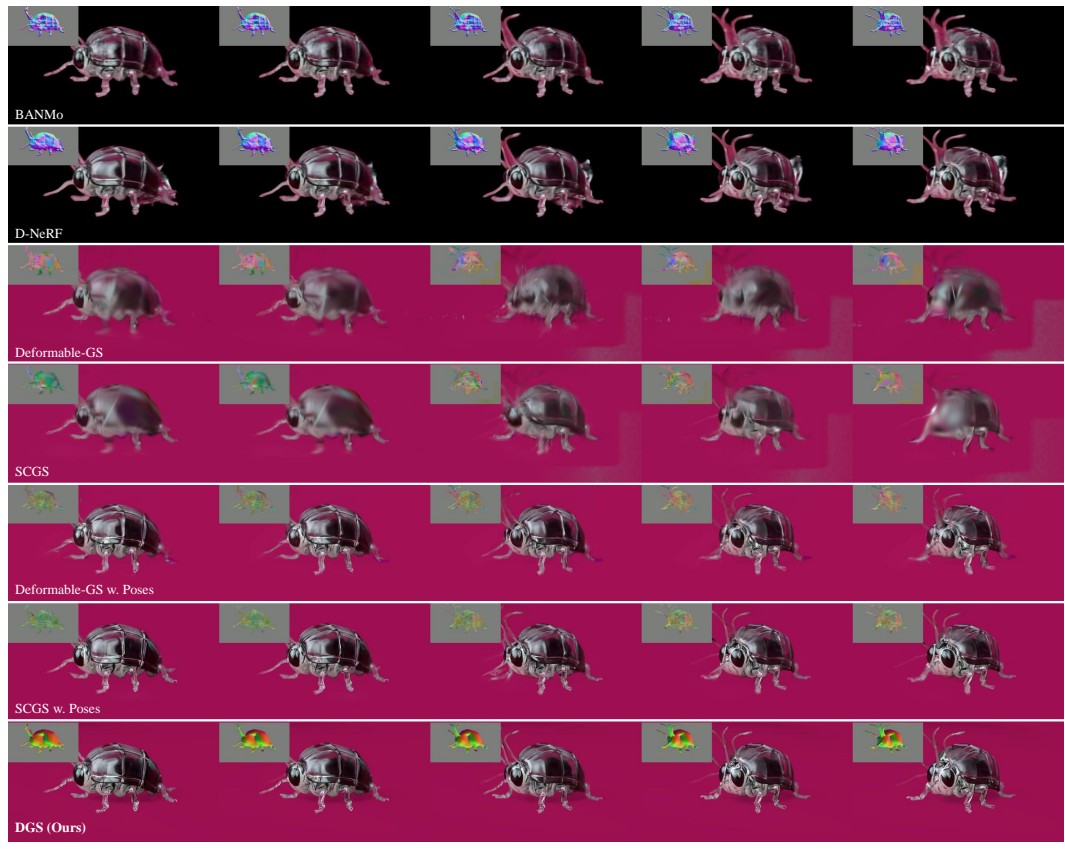

Figure 10: Additional qualitative comparison with more novel views.

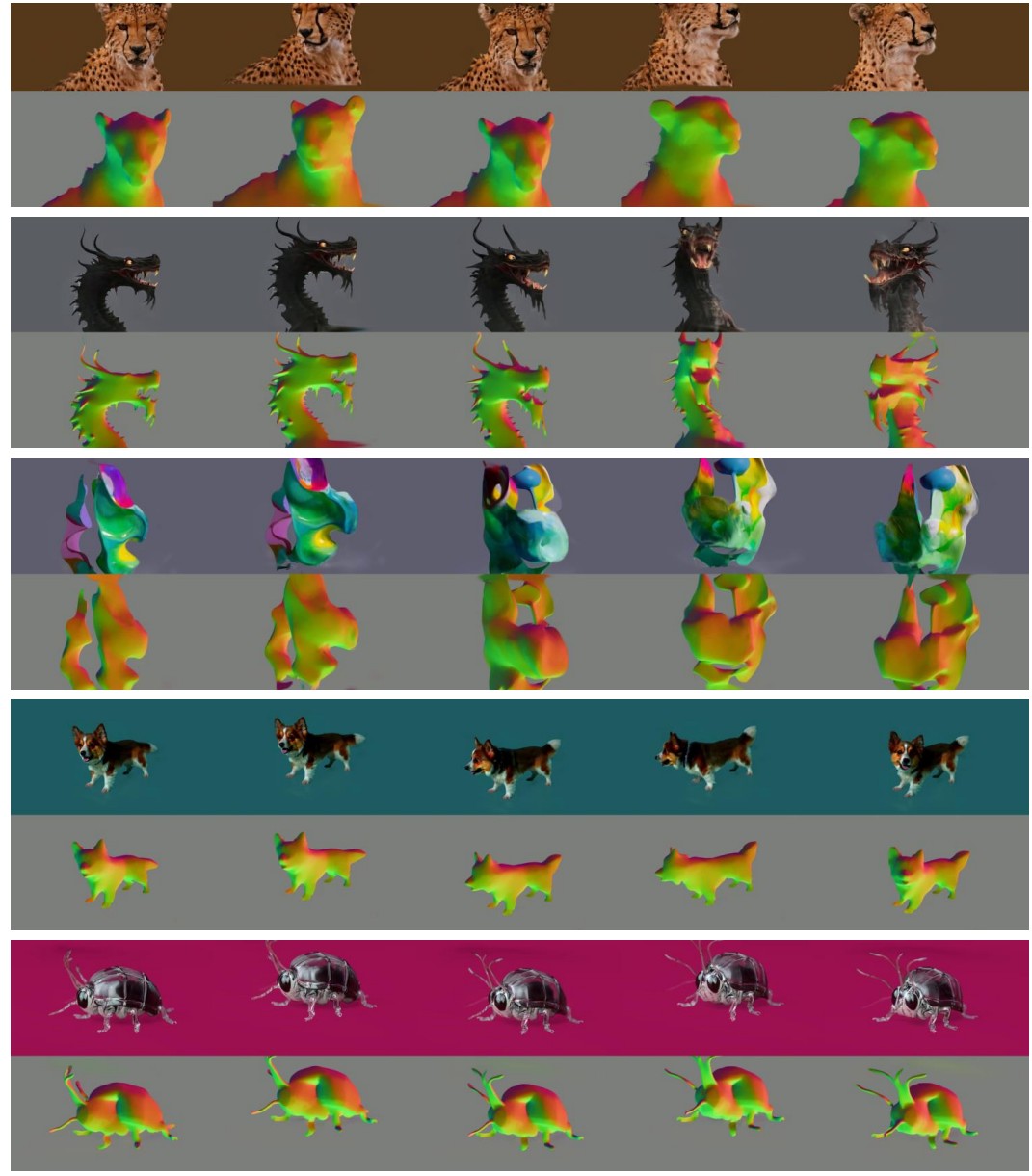

Figure 11: Interpolation on time and views.

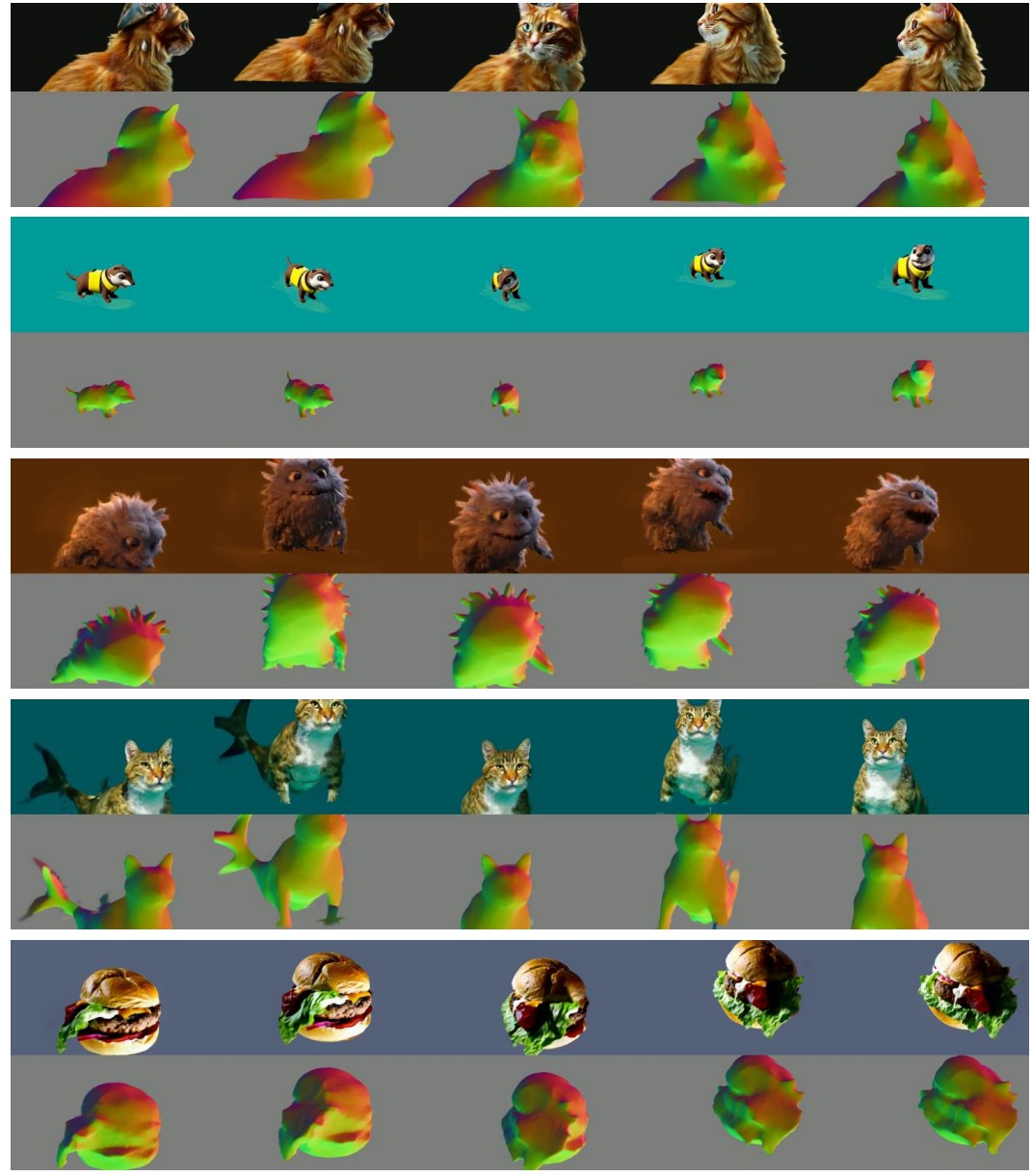

Figure 12: Interpolation on time and views.

