# OpenReview forum: "Vidu4D: Single Generated Video to High-Fidelity 4D Reconstruction with Dynamic Gaussian Surfels"
_NeurIPS.cc/2024/Conference — NeurIPS 2024 poster_

### Official Review · Reviewer_yVYk · 2024-07-11

**Soundness:** 3
**Presentation:** 3
**Contribution:** 3
**Rating:** 5
**Confidence:** 4

**Summary:**

The paper presents Vidu4D, a  reconstruction model that can accurately reconstruct 4D (sequential 3D) representations from single generated videos. This method addressing key challenges and enabling high-fidelity virtual content creation. The proposed techniques, such as Dynamic Gaussian Surfels (DGS) and the initialization state, are good contributions that can benefit the field of multi-modal generation and 4D reconstruction.https://openreview.net/

**Strengths:**

1. This paper is well-written.
2. The qualitative results outperform existing methods.
3. The proposed Dynamic Gaussian Surfels (DGS) approach sounds good. It optimizes time-varying warping functions to transform Gaussian surfels from a static to a dynamically warped state, precisely depicting motion and deformation over time.

**Weaknesses:**

- What is your video foundation model? Is it Stable Video Diffusion, SORA, Open SORA, or Your Vidu?

If you are using an unreleased foundation video model, is the improvement in qualitative results more due to DGS, or is it caused by the video foundation model? If you utilize Vidu, I believe the author needs to provide the results using open source video foundation models like SVD or Open-SORA.

- The quantitative evaluation in the paper is limited to a small set of generated videos.

- The performance of Vidu4D on a more diverse and larger dataset of generated videos is not reported, which could limit the generalizability of the findings.

**Questions:**

The paper does not discuss the computational complexity or runtime performance of Vidu4D, which could be an important consideration for practical applications of the method.

**Limitations:**

The paper does not mention how the Vidu4D  can be extended or adapted to handle other common challenges in 4D reconstruction, such as occlusions, lighting changes, or complex scene dynamics.

---

> ### Author Rebuttal · Authors · 2024-08-07
>
> We sincerely thank you for your valuable comments and insightful suggestions. We address all your comments below. If our response has addressed the concerns, we will highly appreciate it if the reviewer considers raising the score.
>
> **1. Foundation model:** Our foundation model is Vidu. We believe the qualitative results are largely attributed to DGS. We compare the results of other dynamic NeRF and 3DGS in Fig. 4, and our DGS demonstrates state-of-the-art performance.
>
> **2. Open source video foundation models:** For comparison with open-source video generators, we found that generating plausible videos using our specific prompts is challenging for these open-source tools. Instead, we use ToonCrafter to interpolate two frames from our generated video for a fair comparison. It can be observed that, although the generated video lacks consistency, our reconstruction remains relatively stable. Please refer to Fig. I.
>
> **3. More results:** Please refer to Fig. J for results with more styles and categories.
>
> **4. Runtime:** As discussed in Sec. 4.1, for each reconstruction, the overall training takes over 1 hour on an A800 GPU. Specifically, generating a text-guided video takes 4 minutes, preprocessing takes 10 minutes, initialization takes 15 minutes, and the DGS stage takes about 30 minutes.
>
> **5. Challenges in 4D reconstruction:** Please refer to the Motion regularization part in the common response for our discussion about occlusions. For lighting changes, it is promising to introduce a time-dependent spherical harmonic function for Gaussian surfels. To handle complex scene dynamics, such as fluids or flames, adding more control bones or incorporating a dense MLP motion field should be an effective approach.
>
> **6. Benchmark:** Please refer to Q1 in the common response for more quantitative benchmarks.

---

> ### Author Response · Authors · 2024-08-13
>
> Thank you again for your time and effort in reviewing our work and providing the constructive comments. Please feel free to let us know if you have any further questions by August 13 AoE, we are more than happy to address them.

---

> ### Comment · Reviewer_yVYk · 2024-08-13
>
> Thanks for the efforts of the authors for solving some my concerns. I will keep my initial positive rating.

---

### Official Review · Reviewer_7btN · 2024-07-12

**Soundness:** 3
**Presentation:** 1
**Contribution:** 3
**Rating:** 4
**Confidence:** 5

**Summary:**

The paper presents Vidu4D, a  reconstruction model that excels in accurately reconstructing 4D (i.e., sequential 3D) representations from single generated videos, addressing challenges associated with non-rigidity and frame distortion. At the core of Vidu4D is a proposed Dynamic Gaussian Surfels (DGS) technique. DGS optimizes time-varying warping functions to transform Gaussian surfels (surface elements) from a static state to a dynamically warped state. This transformation enables a precise depiction of motion and deformation over time. To preserve the structural integrity of surfacealigned Gaussian surfels, the authors design the warped-state geometric regularization based on continuous warping fields for estimating normals. Additionally, the method learns refinements on rotation and scaling parameters of Gaussian surfels, which alleviates texture flickering during the warping process and enhances the capture of fine-grained appearance details.

**Strengths:**

1. The paper shows interesting visual results, although the anonymous link in the submission pdf seems not working correctly.

2. The paper proposes a Banmo-based dynamic 2dgs formulation for 4d reconstruction.

**Weaknesses:**

1. The paper's annotation is cluttered and extremely hard to follow. Sometimes ignoring some symbols in formulation is desirable when too many of them are presented.

2. The real framework section 3.3 is extremely short. The "more details in our appendix" seems to be a false promise?

3. The model hasn't evaluated on realistic scenes, so that it can be compared with other 4dgs methods using their official results.

4. The paper uses banmo like learnable joint representation to drive deformation, however hasn't mentioned the limitations of this kind of methods, i.e., the 4d scene needs to be object centric.

5. No evaluations of latency, it seems the pipeline needs a dynamic neus/nerf, then init 2dgs at the extracted zero level set, which makes the methods very dependable of the stage one and very time consuming.

**Questions:**

Besides above problems, I think the below questions are needed to be addressed:

What kind of generated video is used as reference video (better to show the reference mono video). Since for 4d reconstruction, if the reference video doesn't show some parts that are occluded across all frames, it is impossible to reconstruct them. An alternative is to use diffusion prior for novel view supervisions to hallucinate these parts, so what is actually happening here????

**Limitations:**

I'm confident it will be horrendously challenging for average readers to fully grip the hole picture without a major revision.
The paper is written to focus on 4d generation task or lift monocular generated video to 4d, however, 80% of the sections are dedicated to representation. If the author want to present this paper as a sknning-based dynamic 2dgs representation paper, then showing 4d generation only is not enough, 4d reconstruction (w/ many widely used benchmarks) should be used as well.

Beside, I hope the author can re-organize the paper no matter it is accepted or not. It is better to put some of the cluttered annotations into appendix, and leave some room for actual pipeline of Vidu4D. The Banmo like joint representation and skinning of 2DGS could be fairly straight forward for people in the field to understand, so no need to put all details in the main paper.

---

> ### Author Rebuttal · Authors · 2024-08-07
>
> We sincerely thank you for your valuable comments and insightful suggestions. We address all your comments below.
>
> **1. Anonymous link:**
> Thank you for your feedback. We believe the issue with the anonymous link might be due to a temporary network problem and we appreciate it if you could try again. Other reviewers might have successfully accessed the videos through the link. If the link remains inaccessible, please refer to Fig. 11-12 in the supplementary and Fig. J for more results.
>
> **2. Annotation and presentation:**
> Thank you for your valuable advice. We acknowledge the need to enhance the clarity of our manuscript. To make our work easier to follow, we will simplify our annotations in future revisions. Additionally, we plan to release our code to aid in the replication and understanding of our research. While reviewers crn6 and yVYk mentioned that our manuscript is well-written and easy to follow, we recognize that there is some room for improvement. We will improve the overall presentation and ensure that our findings are communicated as clearly as possible.
>
> **3. Framework details:**
> In Supp. A, we provide more detail about Vidu4D. Here we also provide more details of initialization and refinement.
>
> In initialization, we first segment the foreground using TrackAnything. Then we extract the unsupervised features with DINOV2, optical flow with VCN [Learning to segment rigid motions from two frames], and metric depth with ZoeDepth. Considering the consistency of the generated video is limited, we register pair-wise camera poses using mask, depth, and flow. Then, we register the root pose of objects using DensePose and initialize the SDF field and warping field with volume rendering. During this process, we also refine the camera poses and root poses using a combination of photometric loss. To enhance registration with unsupervised features, we train an additional channel in NeuS specifically for rendering DinoV2 features, which are then employed for registration purposes as described in RAC. Compared with rasterization, the sampling strategy and continuity of volume rendering make it more suitable for refining poses.
>
> For NeuS rendering, we backward warp sampling points in camera space $\mathbf{X}^t$ to canonical space $\mathbf{X}^*$:
> $$\mathbf{J}^{t, -1} = \mathcal{R}\Big(\sum_{b=1}^{B} w_{b}^{t} \mathcal{Q}(\mathbf{J}^t_{b})^{-1}\Big), $$
> $$\mathbf{X}^t = \mathbf{J}^{t, -1} \mathbf{X}^{*},$$
>
> which is an inversion of Eq. 4 in the main paper. By querying the SDF with $\mathbf{X}^*$, we render RGB and compute the photometric loss to optimize the SDF and the warping field defined in Eq. 6 of the main paper. However, there are two gaps between NeuS warping and DGS warping. First, the sampling points of NeuS are distributed in the frustum of the camera, while the DGS are distributed on the surface. Additionally, we train the inversion of $\mathbf{J}^t$ during initialization, while we utilize the non-inverse ones in DGS. To resolve the distribution gap and ensure that $\mathbf{J}^t$ faithfully models the forward warping, we add a cycle loss:
> $$
> \mathcal{L}_{\mathrm{cyc}} = \|\| \mathbf{J}^t \( \mathbf{J}^{t, -1} \( \mathbf{X}^{t} \) \) - \mathbf{X}^{t} \|\|^{2},
> $$
> where $\mathbf{X}^{t}$ are a combination of mesh surface points and ray sampling points.
>
> After initialization, we extract the mesh in canonical space with the marching cube and initialize Gaussian surfels on the mesh. We set the spherical harmonic in 0-th order to the RGB value of the nearest vertices. Here we keep the warping field and the learned camera poses.
>
> **4. Evaluated on realistic scenes and benchmarks:**
> Please refer to the **common response** for more benchmarks. Specifically, we provide quantitive and qualitative comparisons on realistic scene-level benchmarks (Neural 3D Video dataset and NeRF-DS dataset) and object-level benchmarks (D-NeRF dataset).
>
> **5. Limitation of joint representation:**
> Our method is not limited to reconstructing object-centric scenes. Please see Fig. D and Fig. G in our rebuttal PDF.
>
> **6. Latency:**
> On an Nvidia A800 GPU, generating a 1080p video takes approximately 10 minutes. Preprocessing requires around 12 minutes, initialization takes another 10 minutes, and reconstruction takes 30 minutes. Rendering a 1080p image takes less than 0.1 seconds. We provide rendering latency in Tab. A on the Neural 3D Video benchmark, indicating the superiority of our rendering latency.
>
> **7. Reference video:**
> Due to the submission guidelines, we are unable to upload videos. Instead, we have provided some frames from our reference videos in Fig. K.
>
> **8. Occluded parts:**
> Thank you for your insightful comment. The generative capabilities of Vidu4D are indeed derived from the video generation model. If certain parts of an object are not visible in the reference video, those parts cannot be reconstructed. However, we have found that using the rendered results to guide the video generation model allows for training-free view completion. This approach leverages the strengths of the video generation model to hallucinate and fill in the occluded parts, thereby enhancing the reconstruction process, as shown in Fig. I.
>
> Once again, thank you for your constructive feedback. We'll revise our paper according to your suggestions, move the skinning representation to the supplementary material, and add more details about the Vidu4D pipeline. We will open-source our code to facilitate understanding for readers. We kindly request that you consider raising the score accordingly if we addressed your concerns.

---

> ### Author Response · Authors · 2024-08-13
>
> Thank you again for your time and effort in reviewing our work and providing the constructive comments. Please feel free to let us know if you have any further questions by August 13 AoE, we are more than happy to address them.

---

### Official Review · Reviewer_1Eme · 2024-07-13

**Soundness:** 3
**Presentation:** 2
**Contribution:** 3
**Rating:** 6
**Confidence:** 4

**Summary:**

Video generation models have shown great power recently. Transforming generated videos into 3D/4D representations is important for building a world simulator. This paper proposes an improved 4D reconstruction method from single-generated videos. The key component is the dynamic Gaussian surfels (DGS) technique. Incorporating an initialization stage of a non-rigid warping field, the Vidu4D method produces impressive 4D results with the video generation model Vidu.

**Strengths:**

1. The topic is valuable and interesting to transforming single generated videos into 3D/4D representations. The built 3D/4D representation is more controllable and explicit than a single video. Thus it can be used for rendering more videos with elaborate and customized camera trajectories. Besides, this technique has the potential to be a key component for building a world simulator from video generation models.
2. The provided 4D results show impressive rendering quality, reaching the SOTA performance of this/related field. Besides, the normal looks good, revealing the advantage of modeling geometry from the proposed representation.

**Weaknesses:**

1. The motivation/necessity of building surfels needs to be further strengthened. It is easy to understand building surfels will undoubtedly help reconstruct the surface and geometry. If just considering rendering videos from the built 4D, will a vanilla 4D representation (without improvement on surface reconstruction) be enough? Fig. 4 and Table 1 provide convincing results. However, it is suggested to make it clearer in introducing the motivation, e.g. why better geometry leads to better synthesis.
2. The organization of the method could be improved. For better understanding, it is suggested to first introduce the overall framework of Vidu4D and then demonstrate the dynamic Gaussian Surfels technique.
3. The main method is more like a basic representation/reconstruction approach. Will it still benefit reconstruction from monocular videos captured in real life, not generated videos? Yet, real-life videos have better consistency than generated ones.

**Questions:**

1. The provided results are all object-level 4D ones. Will this method work well on scene-level samples? I know it will be hard to generate the surfels of the background.
2. What about the mesh and depth of the generated 4D representations?
3. One missing related work: Liu I, Su H, Wang X. Dynamic Gaussians Mesh: Consistent Mesh Reconstruction from Monocular Videos[J]. arXiv preprint arXiv:2404.12379, 2024.

**Limitations:**

The paper has clearly addressed the limitations.

---

> ### Author Rebuttal · Authors · 2024-08-07
>
> We sincerely thank you for your valuable comments and insightful suggestions. We address all your comments below. If our response has addressed the concerns and brings new insights to the reviewer, we will highly appreciate it if the reviewer considers raising the score.
>
> **1. Motivation/necessity of building surfels:** Surfels improve the quality of surfaces, which in turn allows for the extraction of high-quality meshes, as illustrated in Fig. F. Furthermore, detailed geometry can enhance downstream applications, such as Gaussian-guided frame continuation at a specific camera pose, as demonstrated in Fig. I of our rebuttal PDF file, where a good depth/normal largely improves the continuation performance.
>
> **2. Organization of the method:** Thank you for the constructive feedback. We will reorganize the methodology section to first introduce the overall framework of Vidu4D, followed by a detailed demonstration of the DGS technique. As discussed in Q2 of the common response, we will provide a more detailed description and analysis of Vidu4D. We hope this adjustment will enhance the clarity and flow of our presentation.
>
> **3. Real monocular videos:** We evaluate our method on realistic scene-level benchmarks (Neural 3D Video dataset and NeRF-DS dataset). Please refer to Tab. C, Tab. D, Fig. D, and Fig. G of the rebuttal PDF file for experiments on real monocular videos.
>
> **4. Meshes and depth:** Please refer to Fig. F for reconstructed meshes and depth.
>
> **5. Missing related work:** Thank you for pointing out the missing related work. We will include a discussion of Liu et al.'s (2024) "Dynamic Gaussians Mesh: Consistent Mesh Reconstruction from Monocular Videos" in our revised manuscript to ensure comprehensive coverage of related research.

---

> > ### Comment · Reviewer_1Eme · 2024-08-13
> >
> > Thanks for the authors' rebuttal! My concerns have been addressed and I will keep my initial positive rating. The supplemented contents are highly recommended to be added to the final version.

---

### Official Review · Reviewer_crn6 · 2024-07-13

**Soundness:** 2
**Presentation:** 3
**Contribution:** 3
**Rating:** 5
**Confidence:** 4

**Summary:**

The paper proposes a technique called Dynamic Gaussian Surfels to effectively reconstruct 4D reqpresentation from a single generated video. DGS optimizes time-varying warping functions to transform Gaussian surfels and the authors adopt Neural SDF for initialization and proposes a geometry regularization technique to preserve the geometry integrity. Extensive experiments on 30 objects proves the effectiveness of the proposed method.

**Strengths:**

a. The resutls are great, show promising application of the proposed DGS.

  b. The paper is well writting and easy to follow.

  c. The authors compare their method with various 4D representations, i.e., skinning and bones, NeRF, Gaussian, and achieve better performance.

**Weaknesses:**

a. The novelty is limited. This work seems to be a combination of LARS (bones and warping), Gaussian Surfels/2DGS (3D representation) and SC-GS/4DGS (refinement). The real novelty might be the geometric regularization and field initialization, although the former is also similar to that used in previous 3D works.

  b. Lack of evaluation details: The authors evaluated the comparison methods on generated video for novel view synthesis in Table 1, however, there is no gt for generated video's novel view resutls. How did the author conduct the evaluation?

  c. The experiments is not extensive enough: The authors claim that their methods are designed for generated video, however, I didn't see any special designs, e.g., sovle the potential multi-view inonconsistency in the input video. So, I think it is a general 4D reconstruction method, and it is recommanded to test the proposed methods on commanly used 4D reconstruction datasets (object and scene-level), using standard evaluation metrics.

**Questions:**

See weaknesses. I might adjust the score according to the response from the authors.

**Limitations:**

Yes

---

> ### Author Rebuttal · Authors · 2024-08-07
>
> We sincerely thank you for your valuable comments and insightful suggestions. We address all your comments below.
>
> **1. Novelty:** To the best of our knowledge, our method is the first to generate 4D content using a text-to-video model. We primarily focus on addressing the spatial-temporal inconsistencies in geometry and texture in generated videos, which is a novel question. Previous works, like SV3D and V3D, focus on enhancing the consistency of video diffusion. This approach is challenging and may lose the knowledge learned from high-quality videos and images when fine-tuning on videos rendered by 3D assets. Instead, we improve the tolerance of the reconstruction method to deal with this inconsistency. Specifically, we absorb multiview inconsistencies into the global motion of bones and unstable textures into the local motion of surfels. We select DGS of a specific frame when we want to obtain statistical meshes or 3DGS. We believe this design is novel.
>
> Our dynamic Gassian surfel is novel. Compared with LARS, we design additional neural skinning weights and a registration technique using DINO features and an invertible warping function. Please refer to Sec. 3.2 and Fig. 6. Compared with Gaussian Surfels/2DGS, we design a warped-state normal regularization to enhance the surface across all frames. Also, we elevate Gaussian Surfels to 3D dimensions for better visual quality, as shown in Fig. 5(c). Compared with SC-GS/4DGS, we applied a relatively simple but plausible warping model containing only 25 anisotropy control bones. This simplification, in contrast to the 500 control points of SC-GS and the dense motion field of 4DGS, helps to regularize motion and prevent overfitting, especially when the camera poses are inaccurate and the object's multi-view consistency cannot be assured, as shown in Fig. B. This method of mitigating Gaussian overfitting is also robust against noise or floater occlusion, as illustrated in Fig. C, where the mask of the dragon covered by sand is missing a piece.
>
> We also design a novel refinement to deal with flickering textures in generated videos. When Gaussian surfels are well reconstructed, its' normal is aligned with the surface normal, and this makes the gradient of density along the surface normal very large:
> $$\mathbf{G}(\mathbf{u})=\text{exp}(-\frac{u^2+v^2}{2}), \frac{d}{dx} \mathbf{G}(\mathbf{u}) = -\left(\frac{1}{\sin^2(\theta)} + \frac{1}{\sin^2(\gamma)}\right) x \exp\left(-\frac{\left(\frac{1}{\sin^2(\theta)} + \frac{1}{\sin^2(\gamma)}\right) x^2}{2}\right)
> $$
> where $\mathbf{G}(\cdot)$ is the density of surfel, $x$ goes along the surface normal, $\theta$ and $\gamma$ are the angles between u, v, and the surface, respectively. Considering that $\theta$ and $\gamma$ are very small, the density is sensitive to the motion along the surface normal. Consequently, this direct of gradient leads the flickering of the texture to be modeled as the position of the surfels moving back and forth over time. When the front-back relationship between the rear surfels and the surface surfels changes, the texture flickers accordingly.
>
> To alleviate this flickering, we introduce a refinement stage that elevates surfels to Gaussian ellipsoids. Compared to surfels, 3D ellipsoids have a smoother density field, and provide a more robust representation during warping, reducing the impact of flickering. In addition, $\Delta \textbf{R}^*_k$ and scaling $\Delta \textbf{S}^*_k$ defined in Eq.8, remove the constraint of aligning the shortest axis with the surface normal. This flexibility makes the 3DGS less likely to unintentionally introduce texture flickering during motion. This is illustrated in Fig. E.
>
> **2. Evaluation details:** We follow the standard evaluation protocol for 4D reconstruction methods such as Hyper-NeRF and SC-GS. Specifically, we build the dataset by using every 4th frame as a training frame and taking the middle frame between each pair of training frames as a validation frame. We will provide additional details in the revised manuscript.
>
>
> **3. Special designs for generated videos:** Please refer to Q2 in the common response for more details.
>
> **4. Experiments on 4D reconstruction datasets:** We follow your advice and add more experiments on the commonly used 4D reconstruction datasets. Please refer to Q2 in the common response for more details.
>
>
> Once again, thank you for your constructive feedback and for considering our paper for acceptance. We'll revise our paper according to your suggestions. We kindly request that you consider raising the score accordingly if we addressed your concerns.

---

> > ### Comment · Reviewer_crn6 · 2024-08-10
> > **Feedback**
> >
> > Thanks for the efforts of the authors for solving my concerns. My feedback is as follows:
> > 1. Novelty:
> >
> > a) This paper is not the first work of generating 4D content using text-to-video models. Previously there are many 4D generation works using text-to-video models, such as 4Dfy (CVPR2024), AYG (CVPR2024), Dream-in-4D(CVPR2024), DG4D (arxiv2023), 4DGen(arxiv2023), aniamte124(arxiv2023), etc. Besides, those works generate 360-degree dynamic objects utilizing text-to-video models. In contrast, this work only generates part of the object, which means the invisible part in the input video is missing in generation results, and the novel view synthesis is limited to small camera movement range.
> >
> > b) Based on the above point, I would suggest the authors to claim that they are the first work which deals with 4D reconstruction from generated videos, rather than the first 4D generation work using text-to-video models.
> >
> >
> > c) I reserve my judgment on the novelty of the 4D reconstruction techniques proposed by the author. I don't think the proposed techniques have special design to handle obvious multi-view inconsistency. (In fact, I didn't see the word "multi-view inconsistency" or similar words in the main paper)
> >
> >
> > 3. Please refer to 1
> >
> > 4. The comparison with SC-GS/D-NeRF/ etc. on dataset w/o gt pose has little meaning. They are not specially designed for no gt scenes. It's suggested to compare with works specially designed for pose-free scenes. For comparison on dataset w/ gt pose, considering there are many regularizations integrated in the proposed pipeline, comparable or slightly better performance is expected.
> >
> > So I decide to maintain the score.

---

> ### Author Response · Authors · 2024-08-12
> **Further response for Reviewer crn6's feedback**
>
> Sincerely thank you for your feedback. We have some further responses w.r.t. (a) 4D reconstruction and (b) special designs for generated videos.
>
> **(a) 4D reconstruction**
>
> Since our focus is on the 4D reconstruction, to the best of our knowledge, there are no **pose-free** benchmarks for **dynamic scenes**. To address your concern, we build the benchmark by collecting openly available videos from the SORA official webpage. We then strictly compare our method against existing state-of-the-art 4D reconstruction methods. We provide the details and results below.
>
> **Benchmark details:** We collect 35 sub-videos from the SORA [1] webpage, including Drone_Ancient_Rome (20 seconds in total, split into 4 sub-videos), Robot_Scene (20 seconds in total, split into 3 sub-videos), Seaside_Aerial_View (20 seconds in total, split into 3 sub-videos), Mountain_Horizontal_View (17 seconds in total, split into 3 sub-videos), Snow_Sakura (17 seconds in total, split into 4 sub-videos), Westworld (25 seconds in total, split into 4 sub-videos), Chrismas_Snowman (17 seconds in total, split into 3 sub-videos), Butterfly_Under_Sea (20 seconds in total, split into 3 sub-videos), Minecraft1 (20 seconds in total, split into 4 sub-videos), Minecraft2 (20 seconds in total, split into 4 sub-videos).
>
> **Evaluation method:** We follow the standard pipeline for dynamic reconstruction (Hyper-NeRF, SC-GS, etc), to construct our evaluation setup by selecting every fourth frame as a training frame and designating the middle frame between each pair of training frames as a validation frame.
>
> **Results:**
>
> | Method | PSNR $\uparrow$ | SSIM $\uparrow$ | LPIPS $\downarrow$ |
> | :---: | :----: | :---: |  :---: |
> | Deformable-GS | 12.72 | 0.5773 | 0.2861 |
> | 4D-GS | 12.15| 0.5609 | 0.2926 |
> | SC-GS | 14.81 | 0.5914 | 0.2420 |
> | SpacetimeGaussians | 13.24 | 0.5836 | 0.2633 |
> | Ours without field initialization (Sec. 3.3) | 15.42 | 0.6167 | 0.2268 |
> | Ours without dual branch refinement (Line 187) | 18.57 | 0.6852 | 0.1945 |
> | **Ours (full model)** | **19.05** | **0.7323** | **0.1839** |
>
>
> Upon acceptance, we will open-source this benchmark and the corresponding codebase to ensure its reproduction. Besides, since currently there are no pose-free benchmarks for dynamic scenes, we believe this built benchmark is a contribution.
>
> Our full model achieves 4.24 PSNR improvement compared to the existing best method. The result proves the superiority of our method and each element we propose (field initialization, dual branch refinement) on the pose-free dynamic scene benchmark.
>
> **(b) Special designs for generated videos**
>
> As we summarized in the common response, properties of generated videos include both larger-scale aspects (unknown poses and unexpected movement) and small-scale aspects (flickering, floater occlusion).
>
> - For larger-scale aspects, we propose the Field initialization stage which provides a proper start for our Dynamic Gaussian Surfels (DGS) regarding both the pose and the movement (please see warping transformation in Eq. 6 of our main paper). Here we'd like to highlight that the field initialization also benefits movement learning since the warping transformation is learned as a continuous field. This design is novel and especially beneficial for generated videos. We will provide more details of the field initialization during revision.
>
> - For small-scale aspects, we have proposed the Dual Branch Refinement (Line 187) and provided ablation studies to prove its effectiveness in alleviating flickering.
>
> Again we are grateful for your feedback.
>
> [1] Video Generation Models as World Simulators.

---

> ### Comment · Reviewer_crn6 · 2024-08-13
>
> Thanks for the author's response. This work might be a pioneering work of pose-free dynamic scene reconstruction. In this case, I would suggest to design a standard and rigor evaluation protocol using real-world multi-view video captured with camera poses. Currently, the evaluation camera pose is perhaps the same as training camera pose, lacking the changes in views, perhaps limited by generated video. I would take it as a benchmark for 4D reconstruction from generated video instead of a benchmark for general pose-free 4D reconstruction methods.
>
> As for the novelty, despite the explanation of the authors, I think this project is without significant technical innovation.
>
>
> But as the first attempt to reconstruct 4D content from generated video, this work is encouraging. Thus, I'll keep my positive score. The authors are suggested to test their work on various video generation models in the future.

---

### Author Rebuttal · Authors · 2024-08-07

We sincerely thank all reviewers's efforts as well as very detailed and insightful suggestions. We find there are common concerns to our paper, and we'd like to clarify them here.

We also add a **PDF file** with more experiment results and visualizations.

_**Q1: From the 4D reconstruction perspective, evaluations on common 4D reconstruction datasets including real scenes and objects? (from crn6, 1Eme, 7btN)**_

Per the reviewers' suggestion, we provide detailed results in our attached PDF file both quantitively and qualitatively, on realistic scene-level benchmarks (Neural 3D Video dataset and NeRF-DS dataset) and object-level benchmarks (D-NeRF dataset). Please See Table A, Table B, Table C, and Table D, and the visualizations in Figure A, Figure D, and Figure G of the PDF file.

**Detailed settings:**
-  We use PSNR, DSSIM, and LPIPS as evaluation metrics and follow the standard setting of SOTA methods to perform training and evaluation. For object-level D-NeRF data and scene-level NeRF-DS data, we train the model for 80000 iterations and start to perform normal regularization on the 40000th iteration. Specifically, on D-NeRF, we perform a group of experiments when ground truth camera poses are unavailable. We train our field initialization stage (Figure 3 of our main paper) for 2000 iterations which takes 10 minutes.

- For the scene-level Neural 3D Video data, we follow the standard setting to train at the resolution $1352\times1014$ with 300 frames per scene. We train the model 30000 iterations with the normal regularization adding from the 10000th iteration.


**Results:**

- Object-level 4D benchmark: From the D-NeRF experimental results in Table A (without GT poses), we observe that our DGS surpasses the second-best method by a large margin (29.06 vs. 20.05 for PSNR).  When given GT poses, our method still outperforms SOTA methods. Our method is especially superior at dynamic normals as shown in Figure A.

- Scene-level (realistic) 4D benchmark: From the Neural 3D Video and NeRF-DS experimental results in Table C and Table D, our DGS achieves the best or the second-best performance in capturing color and shows great superiority in modeling dynamic normals according to the visualizations in Figure D and Figure G.

In summary, for the 4D reconstruction benchmarks, our proposed method demonstrates superior performance compared to existing approaches, particularly in terms of geometric quality and handling unposed scenes.

_**Q2: From the general framework (generated video to 4D) perspective, further strengthened motivation and novelty? (from crn6, 1Eme)**_

**A:** We believe our work has a strong motivation and novelty in terms of the general framework and also special designs for 4D reconstruction from generated videos.

- The overall framework: To the best of our knowledge, our method is the first to generate 4D content using a text-to-video model. The framework of 4D reconstruction from a generated video to achieve 4D generation is novel since it has great potential in modeling high-fidelity 4D representations and many natural downstream applications (as also mentioned by 1Eme). We add a natural example in Figure I of the PDF file that with our framework, we can perform 4D/video customization (based on input text prompts and camera poses) by a training-free video continuation and then render Gaussian.

- Special designs for generated videos: Generated videos and real videos do not have strict boundaries, but usually generated videos are observed to have more unexpected large-scale movement (non-rigidity and distortion leading to complex object root poses) and small-scale anomalies (flickering and float-occlusion). To address both challenges, we design our overall reconstruction method as a coarse-to-fine pipeline.
  -  The coarse part contains time-varying warpings ($\tilde{\textbf{R}}^t$ and $\tilde{\textbf{T}}^t$) to model the basic movement and register the camera and root pose. Besides, we adopt motion regularization (Eq.6) and motion field initialization (Sec. 3.3), together to reconstruct large-scale movement with non-rigidity and distortion even under limited viewpoints. Compared with directly applying 3DGS with dense motion, our design largely alleviates overfitting w.r.t. motions (as shown in Figure B). We provide more details in the response to crn6.
  - The fine part is performed in the static stage using a time-invariant rotation $\Delta \textbf{R}^*_k$ and scaling $\Delta \textbf{S}^*_k$ (Eq.8). This process further alleviates overfitting w.r.t. flickering and float-occlusion (as shown in Figure E and Figure C). We provide more details in the response to crn6.

---

### Decision · Program_Chairs · 2024-09-25

**Decision:**

Accept (poster)

**Comment:**

This paper presents Vidu4D, a reconstruction model that  reconstructs 4D (sequential 3D) representations from single generated videos. The paper received divergent ratings (1x Borderline Reject, 2x Borderline Accept, 1x Weak Accept).

Three reviewers are positive about the method's design and the strong results. However, Reviewer 7btN gave a Borderline Reject rating, citing issues with the writing and organization of the manuscript, as well as some evaluation concerns. The authors' response did not receive a reply. Overall, I agree with the authors that issues with writing clarity should not be grounds for rejection and can be addressed in the revision. I believe this is an interesting paper with strong results that makes a valuable contribution to the area of 4D reconstruction. I will recommend an acceptance.

The authors are requested to revise the paper, taking into account the reviewers' comments for the camera-ready version.